# Learning Clustering-based Prototypes for Compositional Zero-shot Learning

**Hongyu Qu**[1*]**, Jianan Wei**[2*]**, Xiangbo Shu**[1†]**, Wenguan Wang**[2,3]
[1]Nanjing University of Science and Technology    [2]Zhejiang University
[3]National Key Laboratory of Human-Machine Hybrid Augmented Intelligence,
  Xi'an Jiaotong University

## Abstract

Learning primitive (*i.e.*, attribute and object) concepts from seen compositions is the primary challenge of Compositional Zero-Shot Learning (CZSL). Existing CZSL solutions typically rely on oversimplified data assumptions, *e.g.*, modeling each primitive with a single centroid primitive representation, ignoring the natural diversities of the attribute (*resp*. object) when coupled with different objects (*resp*. attribute). In this work, we develop CLUSPRO, a robust clustering-based prototype mining framework for CZSL that defines the conceptual boundaries of primitives through a set of diversified prototypes. Specifically, CLUSPRO conducts within-primitive clustering on the embedding space for automatically discovering and dynamically updating prototypes. These representative prototypes are subsequently used to repaint a well-structured and independent primitive embedding space, ensuring intra-primitive separation and inter-primitive decorrelation through prototype-based contrastive learning and decorrelation learning. Moreover, CLUSPRO efficiently performs prototype clustering in a non-parametric fashion without the introduction of additional learnable parameters or computational budget during testing. Experiments on three benchmarks demonstrate CLUSPRO outperforms various top-leading CZSL solutions under both closed-world and open-world settings. Our code is available at CLUSPRO.

## 1 Introduction

Humans possess the unique ability to recognize a potentially infinite number of novel combinations by associating known components [1], *i.e.*, to make "infinite use of finite means" [2]; for instance, despite never having seen one, people can easily imagine a unicorn by combining their concept of a horse with the idea of a single horn. Inspired by such compositional generalization ability of human intelligence [3, 4], Compositional Zero-Shot Learning (CZSL) [5, 6, 7, 8, 9] is proposed, aiming to recognize unseen attribute-object compositions based on learned knowledge from seen ones.

Existing CZSL solutions [10, 11, 12, 13] typically achieve compositional learning by aligning the visual representation from a pre-trained image encoder backbone with the attribute-object textual representation derived from pre-trained word embeddings. Rather than learning to align visual and textual representation from scratch, recent approaches [14, 15, 16, 17, 18] have pivoted towards leveraging large-scale pre-trained vision-

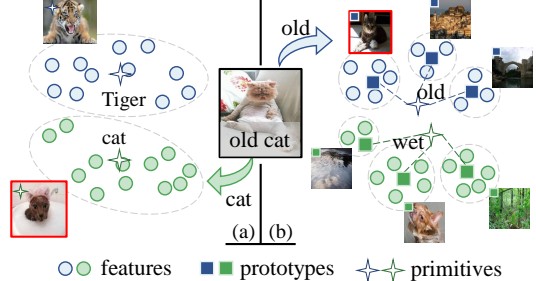

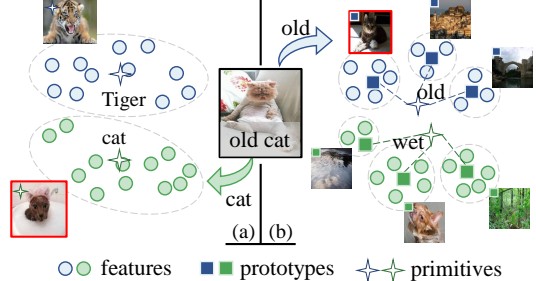

Figure 1: (a) Previous CZSL methods model all samples of each primitive concept with only one centroid primitive presentation, neglecting feature divergence within each primitive when involved in different compositions. (b) Our method represents each primitive as a set of prototypes to capture primitive diversities.

---

*Equal contribution
†Corresponding author

language models (*e.g.*, CLIP[1] [19]) by treating compositional labels as learnable tokens in a pre-defined prompt like "*a photo of [attribute] [object]*". Though impressive, these methods exhibit two limitations: **First**, they struggle to learn visual concepts by modeling an "ideal" primitive (*i.e.*, attribute and object), but ignore an essential issue: each visual concept (*i.e.*, attribute-object pairs) can be semantically similar but visually different. For example, the attribute "broken" combined with "cord" typically signifies disconnection, but conveys the notion of a rugged landscape when applied to "valley". Thus, we argue that a single centroid primitive representation exhibits limited tolerance to intra-primitive variance (Fig. 1a), and it is essential to incorporate more exemplars to capture the natural diversities of primitive. **Second**, they endeavor to explore more effective disentangling strategies (*e.g.*, contrastive learning [20], knowledge distillation [21] or graph representation learning [22]) to achieve independent primitive modeling in a multi-branch manner, but typically present representation disentanglement from a local view (*i.e.*, a few images within a batch [20, 23] or a small training subset [24]), thus failing to perceive underlying data distribution in the entire dataset.

In light of the above, we present CLUSPRO, a clustering-based prototype mining framework for CZSL (Fig. 1b). Specifically, we propose to describe each primitive by abstracting it through a set of representative prototypes, which are automatically discovered by performing within-primitive clustering on the visual representation. Based on these prototypes established across the entire dataset, we introduce two complementary self-supervised learning strategies to repaint the attribute and object embedding spaces, prompting *intra-primitive separation* and *inter-primitive decorrelation*.

Specifically, CLUSPRO employs two disentangling adapters to project visual representation, extracted from a pre-trained image encoder, into separate attribute and object embedding spaces. Then, each primitive is described by clustering $K$ centroids on primitive-wise features. This process (*i.e.*, **Local-aware Prototype Assignment**) involves assigning the visual feature of each primitive to one of a set of prototypes that share the same attribute or object category, while considering the intrinsic coherence of the feature distribution. For computational efficiency, we opt for Generalized Conditional Gradient (GCG) algorithm [25] to enable fast prototype assignment. Additionally, to keep non-learnable prototypes up-to-date, we employ a dynamic **Prototype Updating** mechanism, which recomputes prototypes over the entire dataset in each iteration. The attribute embedding and object embedding, derived from the same visual representation with compositional semantics, inherently exhibit entanglement, which is toxic for prototype construction within primitive. To learn well-structured and independent attribute/object embedding space, we propose two complementary metric learning mechanisms: **i)** *Prototype-based Contrastive Learning* aims to encourage each primitive feature to be similar to its assigned prototype and dissimilar with all other prototypes from the attribute and object branch. In this way, our model can not only capture intra-primitive discriminativeness within the group of attributes or objects, but also promote a clear distinction between attributes and objects. **ii)** *Prototype-based Decorrelation Learning* is devised to shape an independent primitive embedding space (*i.e.*, object representation should be invariant to attribute alterations, and vice versa) by exploring conditional-independence relations between attributes and objects.

CLUSPRO has several appealing merits: **First**, comprehensive modeling of *data distribution*: By conducting within-primitive clustering on the visual embedding space across the entire dataset, CLUSPRO can automatically mine the global data distribution of each primitive from a holistic view. **Second**, explicit supervision of *representation disentanglement*: The clustering-based prototypes enable CLUSPRO to directly shape well-structured yet independent attribute and object embedding spaces via prototype-anchored contrastive learning and decorrelation learning. Such improved primitive embedding spaces, in turn, enable typical within-primitive variation pattern mining. **Third**, high *efficiency*: CLUSPRO perform prototype clustering in a non-parametric fashion without any modification of network architecture or additional computational budget during testing.

To effectively assess our method, we conduct extensive experiments on three gold-standard CZSL datasets (*i.e.*, MIT-States [26], UT-Zappos [27], and C-GQA [28]). Experimental results demonstrate that CLUSPRO significantly exceeds existing state-of-the-arts in both *Close-world (CW)* and *Open-world (OW)* settings (§4.3). Concretely, on the *CW* setting, CLUSPRO achieves **+11.8%** and **+20.2%** AUC gains on UT-Zappos and C-GQA, respectively. On the *OW* settings, it also yields solid improvements of **+19.7%** AUC on UT-Zappos and **+11.1%** AUC on C-GQA. In §4.4, a set of ablative studies confirms the power of our idea and the efficacy of core model designs.

---

[1]Given that CLIP might be exposed to certain unseen compositions during pre-training, we provide detailed data overlap discussion in §G of Appendix.

## 2 RELATED WORK

**Compositional Zero-shot Learning (CZSL).** The goal of CZSL is to recognize unseen attribute-object compositions by combining learned concept knowledge from seen pairs. Early CZSL solutions can be summarized into two paradigms: the first paradigm [29, 28, 5, 30, 31, 32, 33] directly compose attributes and objects with a transformation function and learn a classifier for recognition; the second paradigm [23, 34, 20, 22, 35, 36, 21] mainly decomposes attribute and object in the composition space by well-designed disentangling strategies, *e.g.*, contrastive learning [20], knowledge distillation [21] or graph representation learning [22], and employ two separate classifiers to identify attributes and objects individually. Recent breakthroughs in Vision-Language Models (VLM) [37, 38, 19] make it a promising direction to harness knowledge from pre-trained VLM (*e.g.*, CLIP [19]) for zero-shot and open-vocabulary tasks. Pioneer works [17, 18, 16, 39] build learnable soft prompts with a combined attribute and object vector representation. To capture the contextual nuances in the composition space, recent works [14, 15, 24] jointly model the attribute, object, and composition through vision-language alignments in multiple identification branches.

Despite these advancements, they generally focus on learning one single representative prototype to model each primitive. This limits their ability to interpret the complex and subtle meanings that arise from the combination of various visual concepts. Besides, these methods primarily focus on disentangling attributes and objects with a restricted set of samples, neglecting the potential of incorporating global information to reshape a well-structured and independent embedding space.

**Prototype Learning.** Studies in cognitive psychology evidence that people often explore prototypical knowledge as a foundation for learning and problem-solving across various domains, such as natural language understanding and visual scene understanding [40, 41]. Unlike Softmax-based methods [42, 43, 44], prototype-based classifiers [45, 46, 47, 48] make decisions by computing the distance between new observations and prototype representations of each class. The prototypes typically refer to the centroids of all samples belonging to the same category [49]. For its exemplar-driven nature, a spectrum of recent works attempts to combine deep learning techniques and the idea of prototype learning, boosting great potential in various learning paradigms, including supervised learning [50, 51, 52, 53, 54, 55], few-shot learning [56, 57], and (compositional) zero-shot learning [58, 22, 10]. These (compositional) zero-shot learning works [59, 60, 61, 58] extensively explore prototype learning to enhance feature representation. However, they typically model each class with only one prototype, and their prototypes are often learnable parameters.

Building upon these successes, we aim to advance CZSL by developing a cluster-based prototype learning scheme. Different from previous works [20, 22], which employ one single learnable prototype for each primitive, CLUSPRO explicitly derives prototypes via clustering primitive features over the entire dataset, which are subsequently used to repaint attribute and object embedding spaces.

**Self-supervised Representation Learning.** Self-supervised representation learning (SSRL) methods [62, 63, 64] aim to construct a well-structured embedding space without requiring extensively annotated datasets. Recently, metric learning [65] has emerged as a prominent technique in SSRL, which learns a distance function to reflect the relationships between data points based on their labels. This approach results in more compact, interpretable, and versatile feature representations, which could benefit subsequent tasks, *e.g.*, classification [66, 67, 68] or clustering [69, 70]. It aligns well with CLUSPRO that seeks to automatically discover prototypes of primitive concepts by clustering features associated with coarse-grained labels. Inspired by this, we raise a disentangled representation learning strategy that integrates two complementary self-supervised learning strategies to shape a primitive embedding space with intra-primitive separation and inter-primitive decorrelation.

## 3 METHODOLOGY

### 3.1 PROBLEM STATEMENT

Given the attribute set $\mathcal{A} = \{a_1, a_2, \ldots, a_M\}$ and the object set $\mathcal{O} = \{o_1, o_2, \ldots, o_N\}$, the compositional label set $\mathcal{C}$ can be defined as the Cartesian product between $\mathcal{A}$ and $\mathcal{O}$, *i.e.*, $\mathcal{C} = \mathcal{A} \times \mathcal{O}$. Subsequently, $\mathcal{C}$ is divided into two disjoint subsets: the seen composition set $\mathcal{C}^s$ and the unseen composition set $\mathcal{C}^u$, where $\mathcal{C}^s \cap \mathcal{C}^u = \varnothing$. During training, the model can only access images paired with labels from the seen composition set $\mathcal{C}^s$, *i.e.*, the training set is defined as $\mathcal{T} = \{(x, c) | x \in \mathcal{X}, c \in \mathcal{C}^s\}$,

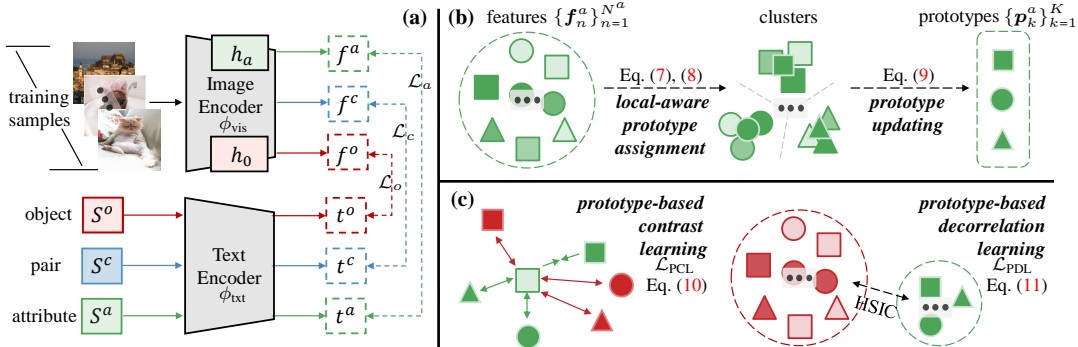

Figure 2: The overview of CLUSPRO. **(a)** CLUSPRO is built upon a three-path paradigm to jointly recognize attribute, object, and attribute-object composition (§3.2). **(b)** To capture the diversity within each primitive, CLUSPRO describes each primitive with a set of prototypes, and conducts within-primitive clustering across training data for prototype assignment and updating (§3.3). **(c)** CLUSPRO imposes two constraints based on these constructed prototypes to promote intra-primitive separation and inter-primitive decorrelation (§3.4).

where $\mathcal{X}$ is the visual space. In the Closed-World (*CW*) setting, the composition space for testing is defined as $\mathcal{C}^t = \mathcal{C}^s \cup \mathcal{C}^u$, where only the known composition space is required. For the Open-World (*OW*) setting, the composition space considers all potential attribute-object pairs, *i.e.*, $\mathcal{C}^t = \mathcal{C}$.

### 3.2 BASELINE ARCHITECTURE

**Encoding Visual Representations.** Our framework is built upon a three-path paradigm [14, 23, 34], which jointly recognizes three kinds of semantic components, *i.e.*, attribute, object, and attribute-object composition. Given an input image $X \in \mathbb{R}^{H \times W \times 3}$, we adopt a visual encoder $\phi^{\text{vis}}$ of CLIP [14] to obtain visual representation $\boldsymbol{f} \in \mathbb{R}^D$. We consider image representation $\boldsymbol{f}$ as composition visual representation $\boldsymbol{f}^c$, and adopt attribute adapter $h^a$ and object adapter $h^o$ [71, 72], each implemented as a separate MLP, to project $\boldsymbol{f}$ into the discriminative attribute and object spaces respectively:

$$\boldsymbol{f}^a = h^a(\boldsymbol{f}), \quad \boldsymbol{f}^o = h^o(\boldsymbol{f}), \quad \boldsymbol{f}^c = \boldsymbol{f}, \tag{1}$$

where $\boldsymbol{f}^a$ and $\boldsymbol{f}^o$ are visual features extracted for attribute and object, respectively.

**Encoding Prompt Representations.** Following existing CZSL [18, 14], we construct prompt representation via a soft learnable prompt strategy for all candidate compositions, attributes, and objects. Specifically, for each attribute-object composition $c_{i,j} = \langle a_i, o_j \rangle$, we create three prompts for each branch, *i.e.*, attribute prompt $\boldsymbol{S}_i^a = [\boldsymbol{s}_1^a, \ldots, \boldsymbol{s}_l^a, \boldsymbol{v}_i^a]$, object prompt $\boldsymbol{S}_j^o = [\boldsymbol{s}_1^o, \ldots, \boldsymbol{s}_l^o, \boldsymbol{v}_j^o]$, and composition prompt $\boldsymbol{S}_{i,j}^c = [\boldsymbol{s}_1^c, \ldots, \boldsymbol{s}_l^c, \boldsymbol{v}_i^a, \boldsymbol{v}_j^o]$, where $\boldsymbol{s}_{1:l}^a$, $\boldsymbol{s}_{1:l}^o$, and $\boldsymbol{s}_{1:l}^c$ are learnable pretix contexts initialized by "*a photo of*". Additionally, $\boldsymbol{v}_i^a$ and $\boldsymbol{v}_j^o$ are trainable vocabulary tokens for the attribute $a_i$ and object $o_j$, respectively. These prompts are then fed into frozen text encoder $\phi^{\text{txt}}$ of CLIP to obtain corresponding prompt features, formulated as:

$$\boldsymbol{t}_i^a = \phi^{\text{txt}}(\boldsymbol{S}_i^a), \quad \boldsymbol{t}_j^o = \phi^{\text{txt}}(\boldsymbol{S}_j^o), \quad \boldsymbol{t}_{i,j}^c = \phi^{\text{txt}}(\boldsymbol{S}_{i,j}^c). \tag{2}$$

**Three-path Learning Objective.** Given visual and prompt representations from three branches, we compute the probabilities for attribute, object, and composition classes, denoted as $p(a_i|\boldsymbol{f}_n)$, $p(o_j|\boldsymbol{f}_n)$, $p(c_{i,j}|\boldsymbol{f}_n)$, respectively. To recognize primitive concepts and their compositions in each branch, three cross-entropy loss functions are employed:

$$\mathcal{L}^a = \frac{1}{N} \sum_{n=1}^N -\log p(a|\boldsymbol{f}_n), \quad p(a_i|\boldsymbol{f}_n) = \frac{\exp(\boldsymbol{f}_n^a \cdot \boldsymbol{t}_i^a / \tau)}{\sum_{k=1}^{|\mathcal{A}|} \exp(\boldsymbol{f}_n^a \cdot \boldsymbol{t}_k^a / \tau)}, \tag{3}$$

$$\mathcal{L}^o = \frac{1}{N} \sum_{n=1}^N -\log p(o|\boldsymbol{f}_n), \quad p(o_j|\boldsymbol{f}_n) = \frac{\exp(\boldsymbol{f}_n^o \cdot \boldsymbol{t}_j^o / \tau)}{\sum_{k=1}^{|\mathcal{O}|} \exp(\boldsymbol{f}_n^o \cdot \boldsymbol{t}_k^o / \tau)}, \tag{4}$$

$$\mathcal{L}^c = \frac{1}{N} \sum_{n=1}^N -\log p(c|\boldsymbol{f}_n), \quad p(c_{i,j}|\boldsymbol{f}_n) = \frac{\exp(\boldsymbol{f}_n^c \cdot \boldsymbol{t}_{i,j}^c / \tau)}{\sum_{k=1}^{|\mathcal{C}|} \exp(\boldsymbol{f}_n^c \cdot \boldsymbol{t}_k^c / \tau)}, \tag{5}$$

where $\tau \in \mathbb{R}$ is pre-defined temperature parameter in CLIP. For simplicity, all the features are $\ell_2$-normalized by default. Then, the three-path classification loss can be formulated as:

$$\mathcal{L}^{\text{BAS}} = \lambda^a \mathcal{L}^a + \lambda^o \mathcal{L}^o + \lambda^c \mathcal{L}^c, \tag{6}$$

where $\lambda^a, \lambda^o, \lambda^c$ are all set to 1, following [14].

**Our Main Idea.** Though impressive, this three-branch paradigm only achieves implicit feature disentanglement to a limited extent by using one single image, failing to perceive the potential structures of the whole dataset. Moreover, it only considers an isolated centroid for each primitive, ignoring rich and diverse intra-primitive patterns. To address this limitation, we propose a clustering-based prototype mining framework (*i.e.*, CLUSPRO), as shown in Fig. 2. Our model not only learns primitive recognition with pre-given semantic labels, but also automatically discovers diverse and fine-grained sub-primitive patterns across the entire dataset. For training, our algorithm alternates between two steps: **i)** perform primitive-wise online clustering to discover sub-primitive prototypes (§3.3); **ii)** impose two prototype-anchored constraints to explicitly shape well-structured and independent attribute/object feature spaces(§3.4). The improved features, in turn, facilitate more reliable primitive-wise clustering, and eventually boost composition predictions.

## 3.3 CLUSTERING-BASED PROTOTYPE MINING

To model the natural diversities of primitives, we exploit rich dataset-level context knowledge to automatically identify informative prototypes within each attribute or object, facilitating primitive concept representation learning. Specifically, we first assign each attribute (*resp.* object) visual feature to the prototypes belonging to the same attribute (*resp.* object) (*i.e.*, **Local-aware Prototype Assignment**), and then continuously update prototypes online according to the assignments (*i.e.*, **Prototype Updating**) with batch training. Such an online clustering strategy forces the model to mine intra-primitive discriminativeness. Notably, we present the online primitive-wise clustering process within both the attribute and object embedding spaces, so as to well represent rich and diverse patterns within each primitive. For clarity, we only explain the prototype construction in the attribute branch, while the object branch follows the same process.

**Local-aware Prototype Assignment.** For each attribute $a \in \mathcal{A}$, we leverage $K$ prototypes ($\{\boldsymbol{p}_k^a\}_{k=1}^K$)[2] to represent its diverse semantic patterns, where $\boldsymbol{p}_k^a$ is $k$-th prototypes of attribute $a$. To get informative yet hidden prototypes, we perform clustering within each attribute on the attribute embedding space. More specifically, for given a set of attribute features $\boldsymbol{F}^a = \{\boldsymbol{f}_n^a\}_{n=1}^{N^a} \in \mathbb{R}^{D \times N^a}$ associated with attribute $a$, where $\boldsymbol{f}_n^a$ is $n$-th attribute features of attribute $a$ and $N^a$ is the number of attribute features, our goal is to assign these attribute features to the $K$ prototypes $\boldsymbol{P}^a = \{\boldsymbol{p}_k^a\}_{k=1}^K \in \mathbb{R}^{D \times K}$. The mapping matrix from $\boldsymbol{F}^a$ to $\boldsymbol{P}^a$ can be denoted as $\boldsymbol{L}^a = [\boldsymbol{l}_n^a]_{n=1}^{N^a} \in \{0, 1\}^{K \times N^a}$, where $\boldsymbol{l}_n^a \in \{0, 1\}^K$ is an one-hot assignment vector of $n$-th attribute features over $K$ prototypes. Let $\boldsymbol{S}^a \in \mathbb{R}^{N^a \times N^a}$ denote cosine similarity among these attribute features $\boldsymbol{F}^a \in \mathbb{R}^{D \times N^a}$ in the attribute embedding space. Thus, the clustering within each attribute can be achieved by the optimization of the assignment matrix $\boldsymbol{L}^a$, *i.e.*, maximizing the similarity $\boldsymbol{Q}^a$ between attribute features $\boldsymbol{F}^a$ and the prototypes $\boldsymbol{P}^a$ (*i.e.*, $\boldsymbol{Q}^a = \mathrm{Softmax}(\boldsymbol{P}^{a\top}\boldsymbol{F}^a) \in \mathbb{R}^{K \times N^a}$), while considering intrinsic coherence structure of features:

$$\min_{\boldsymbol{L}^a \in \mathcal{L}^a} \langle \boldsymbol{L}^{a\top}, -\log \boldsymbol{Q}^a \rangle + \kappa \Omega(\boldsymbol{L}^{a\top}), \tag{7}$$

where $\langle \cdot \rangle$ is the Frobenius dot-product. Note that $\Omega(\boldsymbol{L}^{a\top}) = -\langle \boldsymbol{S}^a, (\boldsymbol{L}^a \odot \boldsymbol{Q}^a)^\top (\boldsymbol{L}^a \odot \boldsymbol{Q}^a) \rangle$ is local coherent regularized term [73], and $\kappa > 0$ is the strength of the regularization, where $\odot$ denotes element-wise multiplication. Different from the classical formulation in [50, 74], *i.e.*, Optimal Transport with entropic constraints, our local-aware prototype assignment can produce superior assignments by fully considering the intrinsic coherence structure of attribute feature distribution, *i.e.*, intra-distribution coherence. Specifically, this term promotes assigning higher weights to $\boldsymbol{L}_{k,i}^a$ and $\boldsymbol{L}_{k,j}^a$ if the $i$-th and $j$-th attribute feature are highly similar (indicated by a high value of $\boldsymbol{S}_{i,j}^a$) and both exhibit a strong similarity, as measured by $\boldsymbol{Q}_{k,i}^a$ and $\boldsymbol{Q}_{k,j}^a$, to the $k$-th prototype of attribute $a$.

As in [69, 75, 76], we relax $\boldsymbol{L}^a$ to an element of transportation polytopes, *i.e.*, $\boldsymbol{L}^a \in \mathbb{R}_+^{K \times N^a}$. Unlike offline clustering [77, 78] requiring multiple passes over the entire dataset, we cast prototype assignment as an optimal transport problem, so as to scale our algorithm to massive data by online clustering:

$$\mathcal{L}^a = \{\boldsymbol{L}^a \in \mathbb{R}_+^{K \times N^a} | \boldsymbol{L}^{a\top}\boldsymbol{1}_K = \boldsymbol{1}_{N^a}, \boldsymbol{L}^a\boldsymbol{1}_{N^a} = \frac{N^a}{K}\boldsymbol{1}_K\}, \tag{8}$$

where $\boldsymbol{1}_K$ denotes the vector of all ones in dimension $K$. $\boldsymbol{L}^{a\top}\boldsymbol{1}_K = \boldsymbol{1}_{N^a}$ is the assignment constraint ensuring each attribute feature is assigned to exactly one prototype, and $\boldsymbol{L}^a\boldsymbol{1}_{N^a} = \frac{N^a}{K}\boldsymbol{1}_K$ is the equipartition constraint, guaranteeing that, on average, each prototype is selected an equal number of

---

[2]For clarity, we slightly reuse $a$ and $o$ to define a certain attribute and object concept, respectively.

times in the batch. With differentiable regularized term and soft assignment relaxation, the solution of Prob. (8) can be given by efficient GCG algorithm [25], which relies on a few matrix-vector multiplications via iterative Dykstra algorithm [79].

**Prototype Updating.** During iterative network training, primitive representations evolve continuously, necessitating offline clustering to recompute sub-primitive prototypes over the entire dataset after each batch, which incurs substantial computational costs. To address this, we propose an *online* clustering approach with momentum updates, where prototypes are dynamically updated using the embeddings within the current mini-batch. In particular, after each training iteration, each prototype $\boldsymbol{p}_k^a$ of the attribute $a \in \mathcal{A}$ is updated as:

$$\boldsymbol{p}_k^a \leftarrow \mu \boldsymbol{p}_k^a + (1 - \mu) \bar{\boldsymbol{f}}_k^a, \tag{9}$$

where $\mu \in [0, 1]$ is a momentum coefficient, and $\bar{\boldsymbol{f}}_k^a \in \mathbb{R}^D$ is the mean vector of the attribute features assigned to the prototype $\boldsymbol{p}_k^a$ by clustering. As such, the prototype updating scheme (Eq. 9) iteratively refines the prototype values in response to the evolving primitive feature representations, thereby facilitating a smoother training process. This online clustering strategy enables our model to effectively discover rich sub-primitive patterns over massive training data.

### 3.4 PROTOTYPE-ANCHORED PRIMITIVE REPRESENTATION LEARNING

By performing online within-primitive clustering separately in the attribute and object embedding spaces, we construct a set of prototypes for each attribute, *i.e.*, $\{\boldsymbol{p}_k^a\}_{k=1}^K$, and each object, *i.e.*, $\{\boldsymbol{p}_k^o\}_{k=1}^K$, to represent diverse sub-primitive patterns. Therefore, the following question naturally arises: *what should a well-structured and independent embedding with discriminative prototypes be like?* To answer this, we enhance the three-path classification loss (Eq. 6) by incorporating two complementary loss constraints based on these constructed prototypes: **Prototype-based Contrastive Learning** and **Prototype-based Decorrelation Learning**, which fully exploits the relationships between primitive features and sub-primitive centers in the embedding space.

**Prototype-based Contrastive Learning.** Our prototype-based contrastive learning strategy contrasts the similarities between each primitive feature, *i.e.*, $\boldsymbol{f}_n \in \boldsymbol{F}^a \cup \boldsymbol{F}^o$, where $\boldsymbol{f}_n$ is $n$-th primitive features, *i.e.*, $\boldsymbol{P}^a \cup \boldsymbol{P}^o$. This strategy encourages each primitive feature $\boldsymbol{f}_n$ to be similar to its assigned prototype $\boldsymbol{p}_+$ and dissimilar to all other $K(M+N)-1$ irrelevant prototypes $\mathcal{P}_-$. Different from only using $K(M+N-1)$ irrelevant prototypes from other primitives as negative samples, our strategy not only ensures inter-primitive separation to some extent, but also guarantee intra-primitive separation. The corresponding training objective for each features $\boldsymbol{f}_n$ is defined as:

$$\mathcal{L}_{\mathrm{PCL}} = -\log \frac{\exp(\boldsymbol{f}_n^\top \cdot \boldsymbol{p}_+ / \tau)}{\exp(\boldsymbol{f}_n^\top \cdot \boldsymbol{p}_+ / \tau) + \sum_{\boldsymbol{p}_- \in \mathcal{P}_-} \exp(\boldsymbol{f}_n^\top \cdot \boldsymbol{p}_- / \tau)}, \tag{10}$$

where $\tau > 0$ is a temperature hyper-parameter. Notably, we treat both attributes and objects equally as primitives to increase the scale and diversity of negative samples.

Our prototype-based contrastive learning exhibits two primary advantages: ❶ Traditional contrastive learning approaches often rely on sophisticated negative sampling strategies to form contrasting pairs, but inevitably yield negative pairs that share similar semantic meaning and should be closer in the embedding space. In contrast, CLUSPRO avoids this long-standing challenge by constructing positive and negative pairs using clustering-based representative prototypes, thus effectively shaping the embedding space by leveraging dataset-level contextual knowledge. ❷ Unlike previous contrastive learning-based CZSL models [20, 80], which necessitate the extra processing for positive and negative feature extraction, our model leverages already constructed prototypes for contrast computation, without incurring extra computational and storage budget.

**Prototype-based Decorrelation Learning.** By leveraging prototype-based contrastive learning, CLUSPRO can effectively distinguish primitive prototypes by maximizing their distance to enhance the independence of linear relationships. But, in CZSL, it is also essential to maintain independence between the embeddings of attributes and objects, *i.e.*, inter-primitive decorrelation; for instance, *apple* should be distinguished from specific attributes, no matter whether *apple* is *red* or *green*. Thus, we propose a prototype-based decorrelation learning to enforce a distinct separation between attribute and object prototypes, ensuring promising disentanglement results. Based on constructed primitive prototypes, the conditional-independence relations can be captured by the following properties: **i)** $\boldsymbol{f}^a \perp\!\!\!\perp \boldsymbol{p}^o$ and **ii)** $\boldsymbol{f}^o \perp\!\!\!\perp \boldsymbol{p}^a$, where $\perp\!\!\!\perp$ denotes the independence between samples. Here, $\boldsymbol{f}^a$ and $\boldsymbol{f}^o$ are disentangled attribute and object features, respectively, with $\boldsymbol{p}^o$ and $\boldsymbol{p}^a$ representing the corresponding sub-primitive prototypes entangled with these features.

Table 1: **Quantitative results**(§4.3) on MIT-States [26], UT-Zappos [27] and C-GQA [28] within *CW* setting.

| *Closed-World* | Backbone | MIT-States | | | | UT-Zappos | | | | C-GQA | | | |
| Method | | Seen↑ | Unseen↑ | HM↑ | AUC↑ | Seen↑ | Unseen↑ | HM↑ | AUC↑ | Seen↑ | Unseen↑ | HM↑ | AUC↑ |
|---|---|---|---|---|---|---|---|---|---|---|---|---|---|
| CLIP [19][ICML2021] | ViT-L | 30.2 | 46.0 | 26.1 | 11.0 | 15.8 | 49.1 | 15.6 | 5.0 | 7.5 | 25.0 | 8.6 | 1.4 |
| CoOp [82][IJCV2022] | ViT-L | 34.4 | 47.6 | 29.8 | 13.5 | 52.1 | 49.3 | 34.6 | 18.8 | 20.5 | 26.8 | 17.1 | 4.4 |
| PCVL [39][Arxiv2022] | ViT-L | 48.5 | 47.2 | 35.3 | 18.3 | 64.4 | 64.0 | 46.1 | 32.2 | - | - | - | - |
| CSP [17][ICLR2023] | ViT-L | 46.6 | 49.9 | 36.3 | 19.4 | 64.2 | 66.2 | 46.6 | 33.0 | 28.8 | 26.8 | 20.5 | 6.2 |
| DFSP(i2t) [18][CVPR2023] | ViT-L | 47.4 | 52.4 | 37.2 | 20.7 | 64.2 | 66.4 | 45.1 | 32.1 | 35.6 | 29.3 | 24.3 | 8.7 |
| DFSP(BiF) [18][CVPR2023] | ViT-L | 47.1 | 52.8 | 37.7 | 20.8 | 63.3 | 69.2 | 47.1 | 33.5 | 36.5 | 32.0 | 26.2 | 9.9 |
| DFSP(t2i) [18][CVPR2023] | ViT-L | 46.9 | 52.0 | 37.3 | 20.6 | 66.7 | 71.7 | 47.2 | 36.0 | 38.2 | 32.0 | 27.1 | 10.5 |
| GIPCOL [83][WACV2024] | ViT-L | 48.5 | 49.6 | 36.6 | 19.9 | 65.0 | 68.5 | 48.8 | 36.2 | 31.9 | 28.4 | 22.5 | 7.1 |
| CDS-CZSL [15][CVPR2024] | ViT-L | 50.3 | 52.9 | 39.2 | 22.4 | 63.9 | 74.8 | 52.7 | 39.5 | 38.3 | 34.2 | 28.1 | 11.1 |
| Troika [14][CVPR2024] | ViT-L | 49.0 | 53.0 | 39.3 | 22.1 | 66.8 | 73.8 | 54.6 | 41.7 | 41.0 | 35.7 | 29.4 | 12.4 |
| PLID [16][ECCV2024] | ViT-L | 49.7 | 52.4 | 39.0 | 22.1 | 67.3 | 68.8 | 52.4 | 38.7 | 38.8 | 33.0 | 27.9 | 11.0 |
| **CLUSPRO (Ours)** | ViT-L | **52.1**±0.6 | **54.0**±0.3 | **40.7**±0.2 | **23.8**±0.2 | **70.7**±1.0 | **76.0**±1.2 | **58.5**±0.6 | **46.6**±0.5 | **44.3**±0.2 | **37.8**±0.2 | **32.8**±0.2 | **14.9**±0.1 |

Specifically, we minimize the correlation between attribute and object embedding spaces by using the Hilbert-Schmidt Independence Criterion (HSIC) [81]. HSIC is a non-parametric, kernel-based statistical measure that evaluates the independence between two continuous random variables, yielding a value of zero if and only if the two variables are statistically independent in the infinite-sample limit. Thus, our prototype-based decorrelation learning strategy can be achieved by minimizing:

$$\mathcal{L}^{\text{PDL}} = \text{HSIC}(\boldsymbol{f}^a, \boldsymbol{p}^o) + \text{HSIC}(\boldsymbol{f}^o, \boldsymbol{p}^a). \tag{11}$$

A similar approach is also adopted by [36]; however, our decorrelation strategy benefits from the use of prototypes, which capture the intrinsic characteristics of primitives, thus more effectively disentangling attribute features and object features in the compositional space.

**Overall Training Objective.** The final learning target of CLUSPRO combines the three-path classification loss $\mathcal{L}^{\text{BAS}}$ (Eq. 6) with prototype-based loss constraints $\mathcal{L}^{\text{PCL}}$ (Eq. 10) and $\mathcal{L}^{\text{PDL}}$ (Eq. 11):

$$\mathcal{L} = \mathcal{L}^{\text{BAS}} + \alpha \mathcal{L}^{\text{PCL}} + \beta \mathcal{L}^{\text{PDL}}, \tag{12}$$

where the coefficients $\alpha$ and $\beta$ are empirically set: $\alpha = 0.2$, $\beta = 0.5$.

**Inference for CSZL.** During testing, the test image $x$ is fed into CLUSPRO to obtain prediction scores for attribute $p(a_i|x)$, object $p(o_i|x)$, and composition prediction $p(c_{i,j}|x)$. Then the final composition class can be predicted by incorporating three branch prediction results:

$$\hat{c} = \arg\max_{c_{i,j} \in \mathcal{C}^t} p(c_{i,j}|x) + p(a_i|x) \cdot p(o_j|x). \tag{13}$$

# 4 EXPERIMENT

## 4.1 EXPERIMENTAL SETUP

**Datasets.** We conduct experiments on three widely-used CZSL benchmarks: MIT-States [26], UT-Zappos [27], and C-GQA [28]. MIT-States consists of $53,753$ natural images in total, with $115$ states and $245$ objects. UT-Zappos contains $50,025$ fine-grain shoe images with $16$ states, $12$ objects and $116$ state-object compositions. C-GQA is the most extensive CZSL dataset, containing $453$ states and $870$ objects for $39,298$ images in total and over $9,500$ state-object compositions. More details are provided in Table 5 (*cf.* §A in Appendix).

**Evaluation Metric.** Following the official evaluation protocol [28, 30, 15, 14], four metrics are adopted for evaluation, *i.e.*, best-Seen accuracy (Seen), best-Unseen accuracy (Unseen), best Harmonic Mean (HM), and Area Under the Curve (AUC). Among them, AUC is the priority as it evaluates the model comprehensively. Please see [84, 30] for full details about metrics.

## 4.2 IMPLEMENTATION DETAILS

**Network Architecture.** CLUSPRO adopts pre-trained CLIP ViT-L/14 model [19], serving as the image and text encoder. The adapters of attribute $h^a$ and object $h^o$ are implemented by two individual MLPs. We group the features of each primitive into $K$ prototypes to describe intra-primitive diversity. The number of prototypes $K$ and the momentum coefficient $\mu$ in Eq. 9 are empirically set to 5 and 0.99, respectively (ablation study in Table 4a and 4b). We follow [73] to set $\kappa = 1$ in Eq. 7.

**Training.** CLUSPRO is trained end-to-end for 15 epochs with Adam optimizer [85]. To manage the learning rate, we initialize it at $1e-4$ for all datasets and set weight decay to $5e-5$. The coefficients

Table 2: **Quantitative results**(§4.3) on MIT-States [26], UT-Zappos [27] and C-GQA [28] within *OW* setting .

| *Open-World* Method | Backbone | MIT-States | | | | UT-Zappos | | | | C-GQA | | | |
|---|---|---|---|---|---|---|---|---|---|---|---|---|---|
| | | Seen↑ | Unseen↑ | HM↑ | AUC↑ | Seen↑ | Unseen↑ | HM↑ | AUC↑ | Seen↑ | Unseen↑ | HM↑ | AUC↑ |
| CLIP [19][ICML2021] | ViT-L | 30.1 | 14.3 | 12.8 | 3.0 | 15.7 | 20.6 | 11.2 | 2.2 | 7.5 | 4.6 | 4.0 | 0.3 |
| CoOp [82][IJCV2022] | ViT-L | 34.6 | 9.3 | 12.3 | 2.8 | 52.1 | 31.5 | 28.9 | 13.2 | 21.0 | 4.6 | 5.5 | 0.7 |
| PCVL [39][Arxiv2021] | ViT-L | 48.5 | 16.0 | 17.7 | 6.1 | 64.6 | 44.0 | 37.1 | 21.6 | - | - | - | - |
| CSP [17][ICLR2023] | ViT-L | 46.3 | 15.7 | 17.4 | 5.7 | 64.1 | 44.1 | 38.9 | 22.7 | 28.7 | 5.2 | 6.9 | 1.2 |
| DFSP(i2t) [18][CVPR2023] | ViT-L | 47.2 | 18.2 | 19.1 | 6.7 | 64.3 | 53.8 | 41.2 | 26.4 | 35.6 | 6.5 | 9.0 | 2.0 |
| DFSP(BiF) [18][CVPR2023] | ViT-L | 47.1 | 18.1 | 19.2 | 6.7 | 63.5 | 57.2 | 42.7 | 27.6 | 36.4 | 7.6 | 10.6 | 2.4 |
| DFSP(t2i) [18][CVPR2023] | ViT-L | 47.5 | 18.5 | 19.3 | 6.8 | 66.8 | 60.0 | 44.0 | 30.3 | 38.3 | 7.2 | 10.4 | 2.4 |
| GIPCOL [83][WACV2024] | ViT-L | 48.5 | 16.0 | 17.9 | 6.3 | 65.0 | 45.0 | 40.1 | 23.5 | 31.6 | 5.5 | 7.3 | 1.3 |
| CDS-CZSL [15][CVPR2024] | ViT-L | 49.4 | 21.8 | 22.1 | 8.5 | 64.7 | 61.3 | 48.2 | 32.3 | 37.6 | 8.2 | 11.6 | 2.7 |
| Troika [14][CVPR2024] | ViT-L | 48.8 | 18.7 | 20.1 | 7.2 | 66.4 | 61.2 | 47.8 | 33.0 | 40.8 | 7.9 | 10.9 | 2.7 |
| PLID [16][ECCV2024] | ViT-L | 49.1 | 18.7 | 20.0 | 7.3 | 67.6 | 55.5 | 46.6 | 30.8 | 39.1 | 7.5 | 10.6 | 2.5 |
| **CLUSPRO (Ours)** | ViT-L | **51.2**$_{\pm0.4}$ | **22.1**$_{\pm0.2}$ | **23.0**$_{\pm0.1}$ | **9.3**$_{\pm0.2}$ | **71.0**$_{\pm1.1}$ | **66.2**$_{\pm1.0}$ | **54.1**$_{\pm0.7}$ | **39.5**$_{\pm0.8}$ | **41.6**$_{\pm0.3}$ | **8.3**$_{\pm0.2}$ | **11.6**$_{\pm0.3}$ | **3.0**$_{\pm0.1}$ |

$\alpha$ and $\beta$ in overall training objective (Eq. 12) are empirically set to 0.2 and 0.5, respectively (related experiments in §D of Appendix). In Eq. 10, the temperature parameter $\tau$ is maintained at 0.1. CLUSPRO is implemented in PyTorch and trained on one NVIDIA RTX 4090 GPU.

**Testing.** Following previous works [17, 83], we apply the post-training calibration to filter out infeasible compositions in the open-world setting during testing. Note that, during model deployment, there is no any network architectural modification or extra inference cost introduced to the base model. The primitive prototypes, $\boldsymbol{P}^a$ and $\boldsymbol{P}^o$, are directly discarded after network training.

## 4.3 COMPARISON TO STATE-OF-THE-ARTS

In this section, we compare our method CLUSPRO with top-leading CZSL solutions on three dataset (*i.e.*, MIT-States [26], UT-Zappos [27], and C-GQA [28]) in *CW* and *OW* settings.

**Performance on *CW* Setting.** As summarized in Table 1, our approach CLUSPRO outperforms recent state-of-the-art (SOTA) CZSL algorithms across all datasets [26, 27, 28] on *CW* setting. Concretely, in terms of AUC which is the priority metric for evaluating the model comprehensively, CLUSPRO yields **+1.4**, **+4.9**, and **+2.5** AUC score gains compared with SOTA methods on MIT-States, UT-Zappos, and C-GQA, respectively. Besides, CLUSPRO boosts HM to **40.7** (**+3.6**%) on MIT-States, **58.5** (**+7.1**%) on UT-Zappos, and **32.8** (**+11.6**%) on C-GQA. Moreover, CLUSPRO earns consistent best Seen Accuracy (Seen) and Unseen Accuracy (Unseen) improvement. These consistency improvements are attributed to the fact that our algorithm captures diverse sub-primitive patterns, *i.e.*, intra-primitive variations, which improves generalization on unseen compositions.

**Performance on *OW* Setting.** Table 2 reports comparison results on *OW* setting. As seen, most CZSL methods suffer a substantial performance drop due to vast search space in *OW* setting. In contrast, our method CLUSPRO still surpasses all published competitors across three datasets [26, 27, 28]. In particular, CLUSPRO attains the highest AUC scores: **9.3** (**+9.4**%) on MIT-States, **39.5** (**+19.7**%) on UT-Zappos, and **3.0** (**+11.1**%) on C-GQA. In terms of HM, best Seen Accuracy (Seen) and Unseen Accuracy (Unseen), CLUSPRO still achieves the best results. This reinforces our belief that learning a group of discriminative prototypes for each primitive helps our model to recognize unseen compositions, even within challenging *OW* setting.

## 4.4 DIAGNOSTIC EXPERIMENT

To evaluate our algorithm designs and gain further insights, we conduct ablation studies on UT-Zappos [26] and C-GQA [28] in *CW* settings.

**Key Component Analysis.** We first investigate the effectiveness of our core idea, *i.e.*, clustering-based prototype learning. To make use of discovered rich sub-primitive prototypes to shape attribute and object embedding spaces, two key training objectives are proposed, *i.e.*, Prototype-based Contrast $\mathcal{L}^{PCL}$ (Eq. 10) and Decorrelation $\mathcal{L}^{PDL}$ (Eq. 11). As shown in Table 3, we build BASELINE that trains in the three-branch paradigm, without within-primitive prototype clustering (*i.e.*, prototype assignment and updating). We can find that, adding $\mathcal{L}^{PCL}$ or $\mathcal{L}^{PDL}$ individually leads to a substantial performance gain, *e.g.*, **+4.9/+2.8** AUC on UT-Zappos [26], and **+2.4/+1.9** AUC on C-GQA [28]. This verifies the efficacy of explicitly promoting inter-primitive prototype separation and attribute-object independence. Last, by combining $\mathcal{L}^{PCL}$ and $\mathcal{L}^{PDL}$, our full model yields the best results.

Table 3: **Analysis of core components** (§4.4) on UT-Zappos [27] and C-GQA [28] within **CW** setting.

| | $\mathcal{L}_{PCL}$ (Eq.10) | $\mathcal{L}_{PDL}$ (Eq.11) | UT-Zappos Seen↑ | Unseen↑ | HM↑ | AUC↑ | C-GQA Seen↑ | Unseen↑ | HM↑ | AUC↑ |
|---|---|---|---|---|---|---|---|---|---|---|
| BASELINE (*w/o* Clustering) | | | 66.2 | 74.6 | 54.1 | 41.0 | 40.5 | 33.4 | 29.7 | 11.8 |
| Prototype-bsaed Contrast | ✓ | | 69.9 | 74.7 | 57.1 | 45.9 | 43.6 | 36.7 | 32.1 | 14.2 |
| Prototype-bsaed Decorrelation | | ✓ | 67.6 | 75.2 | 56.5 | 43.8 | 43.1 | 36.1 | 31.4 | 13.7 |
| Contrast + Decorrelation | ✓ | ✓ | **70.7** | **76.0** | **58.5** | **46.6** | **44.3** | **37.8** | **32.8** | **14.9** |

Table 4: **A set of ablation studies** on UT-Zappos [27] (§4.4).

| Prototype $K$ | UT-Zappos Seen↑ | Unseen↑ | HM↑ | AUC↑ |
|---|---|---|---|---|
| $K = 1$ | 68.9 | 73.6 | 55.2 | 42.8 |
| $K = 3$ | 70.6 | 74.6 | 56.9 | 45.1 |
| $K = 5$ | **70.7** | **76.0** | **58.5** | **46.6** |
| $K = 10$ | 69.9 | 75.4 | 58.3 | 46.0 |
| $K = 20$ | 70.1 | 75.2 | 58.0 | 45.9 |

(a) Per-primitive Prototype Number

| Coefficient $\mu$ | UT-Zappos Seen↑ | Unseen↑ | HM↑ | AUC↑ |
|---|---|---|---|---|
| $\mu = 0$ | 67.2 | 74.1 | 54.9 | 42.6 |
| $\mu = 0.5$ | 69.8 | 75.1 | 56.0 | 43.3 |
| $\mu = 0.9$ | 70.5 | 74.9 | 58.1 | 46.0 |
| $\mu = 0.99$ | **70.7** | **76.0** | **58.5** | **46.6** |
| $\mu = 0.999$ | 70.5 | 74.7 | 57.0 | 45.1 |

(b) Prototype Updating Coefficient $\mu$

| Clustering Branch Attribute | Object | UT-Zappos Seen↑ | Unseen↑ | HM↑ | AUC↑ |
|---|---|---|---|---|---|
| | | 66.2 | 74.6 | 54.1 | 41.0 |
| ✓ | | 69.7 | 74.3 | 56.5 | 44.3 |
| | ✓ | 69.5 | 73.6 | 56.4 | 44.1 |
| ✓ | ✓ | **70.7** | **76.0** | **58.5** | **46.6** |

(c) Clustering Branch

| Clustering Strategy | UT-Zappos Seen↑ | Unseen↑ | HM↑ | AUC↑ |
|---|---|---|---|---|
| None | 66.2 | 74.6 | 54.1 | 41.0 |
| Cosine Similarity | 68.9 | 74.7 | 57.7 | 45.0 |
| Classical OT | 70.1 | 75.2 | 58.2 | 45.8 |
| Ours | **70.7** | **76.0** | **58.5** | **46.6** |

(d) Clustering Strategy

**Prototype Number Per Primitive $K$.** We next investigate the impact of the prototype number per primitive. The results are reported in Table 4a. Note that for $K = 1$, each primitive is directly represented by the mean embedding of primitive features in the current batch without prototype clustering. As shown in Table 4a, this baseline yields the HM score of 55.2 and AUC score of 42.8 on UT-Zappos [26], respectively. When representing one primitive concept with a group of prototypes, we observe that our model CLUSPRO gains stable improvements (*i.e.*, HM: 55.2→**58.5**, AUC: 42.8→**46.6**) as the number of prototypes grows (*i.e.*, $K = 5$). This supports our hypothesis that leveraging a set of diversified prototypes to describe a primitive concept can capture diverse intra-primitive patterns. However, too many prototypes above $K = 5$ results in negative gains. This may be because CLUSPRO suffers from insignificant sub-primitive patterns produced by over-clustering.

**Momentum Coefficient $\mu$.** Table 4b probes the impact of momentum coefficient $\mu$ (Eq.9), which controls the speed of primitive prototype updating. We can clearly observe that, our algorithm performs better with a relatively large coefficient (*i.e.*, $\mu = 0.99$), verifying that slow updating is beneficial, but not too slow (*i.e.*, $\mu = 0.999$). When $\mu$ is too small, the performance decreases. In particular, our algorithm encounters a large decrease at the extreme of no momentum (*i.e.*, $\mu = 0$).

**Multi-branch Clustering.** In Table 4c, we study the impact of attribute and object clustering branches by removing one or more specific branches. Removing one branch clustering means that, we discard primitive-wise clustering to mine sub-primitive patterns in the corresponding branch, and remove the corresponding training objectives from Eq. 10 and 11. As shown in Table 4c, using attribute or object clustering branches individually only yields limited performance gains, *e.g.*, **+3.3/+3.1** AUC score. By unifying the two clustering branches together, our full model achieves the best performance across four metrics, confirming their complementarity.

**Clustering Strategy.** We examine the impact of our proposed local-aware clustering strategy (*cf.* Eq.7) by contrasting it with the model without primitive-wise clustering, the cosine similarity updating [86], and classical Optional Transport (OT) [87, 88]. As shown in Table 4d, our local-aware clustering strategy proves to be more effective: it outperforms the model without clustering, the cosine similarity, and classical OT across all metrics, *e.g.*, **+4.7**, **+1.6**, and **+0.8** AUC scores on UT-Zappos, respectively. This study confirms that considering the intrinsic coherence structure of attribute/object feature distribution is beneficial for superior prototype assignment.

## 4.5 QUALITY ANALYSIS

**Success Cases.** The first four columns of Fig. 3 present success cases of our method CLUSPRO for both seen and unseen compositions on UT-Zappos [27] and C-GQA [28]. As seen, compared with the base model without primitive-wise prototype clustering, CLUSPRO works much better. Even for the complex C-GQA dataset, CLUSPRO still correctly predicts labels. For example, CLUSPRO can

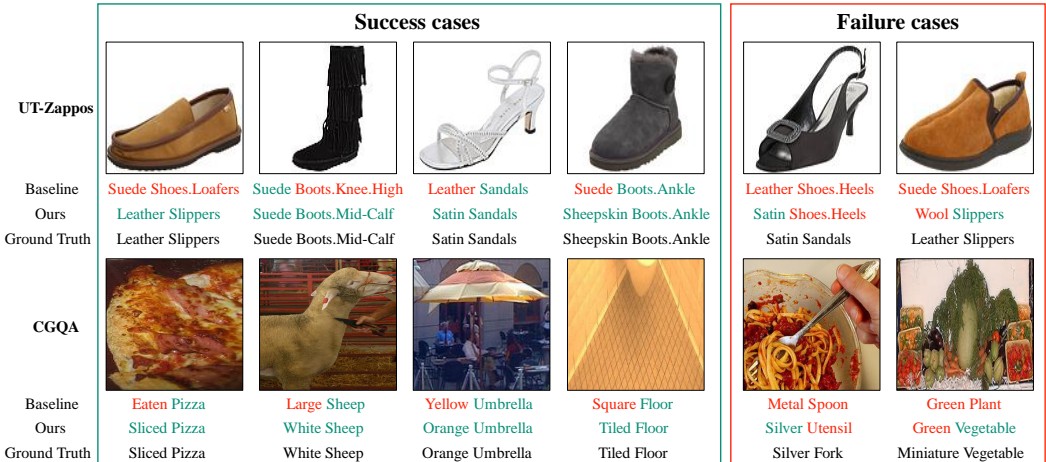

Figure 3: Case study on UT-Zappos [27] and C-GQA [28]. We compare CLUSPRO with baseline without primitive-wise prototype clustering. Correct and incorrect predictions are marked in **green** and **red**, respectively.

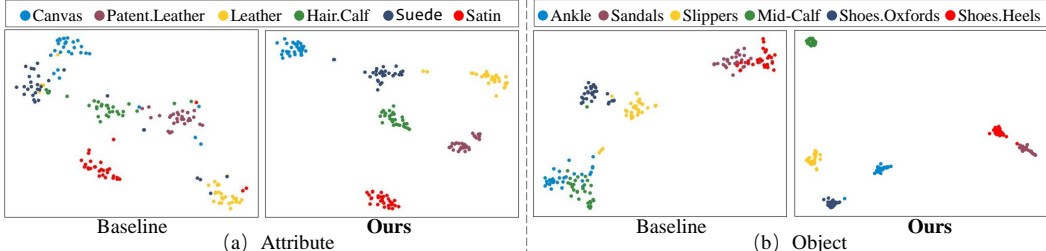

Figure 4: Visualization of attribute and object features learned by baseline and CLUSPRO on UT-Zappos [27].

calibrate *Suede* to *Leather* (materials) and *Yellow* to *Orange* (colors). This demonstrates CLUSPRO can capture fine-grained primitive patterns (*e.g.*, various materials and colors) by representing each primitive as a set of prototypes. Moreover, benefiting from prototype clustering across the whole dataset, CLUSPRO automatically mines the global data distribution of each primitive, leading to generalizing well to unseen compositions. More success cases are provided in §E of Appendix.

**Failure Cases and Limitations.** The last two columns of Fig. 3 show failure cases, where the attribute and object of images are highly entangled and visually confusing. However, CLUSPRO still identifies the part of attribute-object compositions. In addition, though making mistakes on attribute predictions (*e.g.*, *Green Vegetable* in column 6, row 2), such wrong predictions interpret another attribute (*i.e.*, the color) of *Miniature Vegetable*. Thus we will make use of large language models to generate informative descriptions for each composition in the future, so as to emphasize primary primitives. More failure cases are provided in §E of Appendix.

**Feature Distributions of Attribute and Object.** We visualize learned features of attribute and object by $\mathcal{L}_{BAS}$ (Eq. 6) and $\mathcal{L}$ (Eq. 12) in Fig. 4. We observe that, after considering clustering-based prototype mining, learned attribute and object features become more compact and better separated. This demonstrates that CLUSPRO can shape well-structured attribute/object embedding spaces by clustering-based analysis across the whole dataset, hence ensuring better visual disentanglement.

## 5 CONCLUSION

In this work, we present CLUSPRO, a clustering-based prototype mining framework for Compositional Zero-Shot Learning. This framework aims to learn a well-structured and independent embedding space with multiple discriminative prototypes for each primitive, which alternates between two steps: 1) within-primitive online clustering for automatically discovering and dynamically updating prototypes; 2) prototype-based primitive representation learning for promoting intra-primitive separation and inter-primitive decorrelation. Experimental results on three gold-standard datasets demonstrate the superiority of our clustering-based scheme against existing methods.

## 6 ACKNOWLEDGEMENT

This work was supported by the National Natural Science Foundation of China (No. 62222207, 62332010, 62427808, and 62372405), the Fundamental Research Funds for the Central Universities 226-2024-00058, the National Key Laboratory of Human-Machine Hybrid Augmented Intelligence, Xi'an Jiaotong University (No. HMHAI-202403), Bytedance Doubao Fund, and Earth System Big Data Platform of the School of Earth Sciences, Zhejiang University.

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

- §A Detailed data split statistics.

- §B Algorithm overview.

- §C More details about baseline model.

- §D More quantitative results.

- §E More qualitative visualization.

- §F The pseudo-code of prototype assignment and updating.

- §G More discussions.

## A    DETAILED DATA SPLIT STATISTICS

We conduct experiments on three widely-used CZSL benchmarks: MIT-States [26], UT-Zappos [27], and C-GQA [28]. MIT-States consists of $53,753$ natural images in total, with $115$ states and $245$ objects. Following conventional procedures, $1,962$ available compositions in the dataset are split into $1,262$ seen and $300/400$ unseen compositions for `train`/`validation`/`test`, respectively. UT-Zappos contains $50,025$ fine-grain shoe images with $16$ states, $12$ objects and $116$ state-object compositions. Following standard practices, the compositions are split into $83$ seen compositions, $15$ seen and $15$ unseen compositions, $18$ seen and $18$ unseen compositions for `train`/`validation`/`test` splits. C-GQA is the most extensive CZSL dataset, containing $453$ states and $870$ objects for $39,298$ images in total and over $9,500$ state-object compositions. The dataset is divided into $5,592$ seen compositions for `train`, $1,252$ seen and $1,040$ unseen compositions for `validation`, and $888$ and $923$ unseen compositions for `test`. The detailed data split statistics is provided in Table 5. Here $|\mathcal{C}^s|$ and $|\mathcal{C}^u|$ indicate the number of seen and unseen compositions, respectively. $|\mathcal{X}|$ represents the number of images.

Table 5: The detailed data split statistics (§A) on MIT-States [26], UT-Zappos [27] and C-GQA [28].

| Dataset | $|\mathcal{A}|$ | $|\mathcal{O}|$ | train $|\mathcal{C}^s|$ | $|\mathcal{X}|$ | validation $|\mathcal{C}^s|$ | $|\mathcal{C}^u|$ | $|\mathcal{X}|$ | test $|\mathcal{C}^s|$ | $|\mathcal{C}^u|$ | $|\mathcal{X}|$ |
|---|---|---|---|---|---|---|---|---|---|---|
| MIT-States [26] | 115 | 245 | 1262 | 30k | 300 | 300 | 10k | 400 | 400 | 13k |
| UT-Zappos [27] | 16 | 12 | 83 | 23k | 15 | 15 | 3k | 18 | 18 | 3k |
| C-GQA [28] | 413 | 674 | 5592 | 27k | 1252 | 1040 | 7k | 888 | 923 | 5k |

## B    ALGORITHM OVERVIEW

Fig. 2 presents the architecture of our CLUSPRO. It takes a batch of images and all the semantic labels (*i.e.*, attributes, objects, and compositions) as input. CLUSPRO first utilizes the visual encoder of CLIP [19] along with attribute and object adapters to obtain attribute features $F^a$, object features $F^o$, and composition features $F$ (Eq. 1). Besides, CLUSPRO constructs attribute, object, and composition prompt representations (Eq. 2) via a soft learnable prompt strategy [14] based on pre-given semantic labels. Based on these visual and prompt representations from three branches, the three-path classification loss (*i.e.*, $\mathcal{L}^{\text{BAS}}$) in Eq. 6 are employed to recognize primitive concepts and their compositions. Meanwhile, based on the obtained attribute and object visual feature representation (*i.e.*, $F^a$ and $F^o$), CLUSPRO learn prototypes by within-primitive clustering (§3.3) and then propose two complementary metric learning mechanisms (*i.e.*, $\mathcal{L}^{\text{PCL}}$ and $\mathcal{L}^{\text{PDL}}$) based on these prototypes, so as to explicitly shape well-structured and independent primitive embedding space (§3.4). Finally, we assemble the three-path classification loss $\mathcal{L}^{\text{BAS}}$ and our proposed prototype-based loss constraints (*i.e.*, $\mathcal{L}^{\text{PCL}}$ and $\mathcal{L}^{\text{PDL}}$) as our final learning objective. Our algorithm not only learns primitive recognition with pre-given semantic labels, but also automatically discovers diverse and fine-grained intra-primitive patterns via a set of prototypes across the entire dataset.

# C  MORE DETAILS ABOUT BASELINE MODEL

**Visual Feature Extraction.** Following [14, 15, 24], we adopt the visual encoder $\phi^{\text{vis}}$ of CLIP [19] to splits the input image $X \in \mathbb{R}^{H \times W \times 3}$ into $N_p = HW/P$ patches, where $P$ is the resolution of each patch. Note that we following [14, 24] to tune the image encoder of CLIP with LoRA [71, 72], a lightweight parameter efficient fine-tuning (PEFT) strategy. The encoder $\phi^{\text{vis}}$ projects these patches into patch tokens along with a [cls] token, and then updates these tokens via Transformer blocks. Finally, the [cls] token serves as the image representation $\boldsymbol{f}^c$. We adopt attribute adapter $h^a$ and object adapter $h^o$ [71, 72], each implemented as a separate MLP, to project $\boldsymbol{f}^c$ into the discriminative attribute feature $\boldsymbol{f}^a$ and object feature $\boldsymbol{f}^o$, respectively.

**Prompt Feature Extraction.** We follow existing CZSL [18, 14] to employ an independent prompt prefix for each branch. Specifically, for each attribute-object composition $c_{i,j} = \langle a_i, o_j \rangle$, we create three prompts for each branch, *i.e.*, attribute prompt $\boldsymbol{S}_i^a = [\boldsymbol{s}_1^a, \ldots, \boldsymbol{s}_l^a, \boldsymbol{v}_i^a]$, object prompt $\boldsymbol{S}_j^o = [\boldsymbol{s}_1^o, \ldots, \boldsymbol{s}_l^o, \boldsymbol{v}_j^o]$, and composition prompt $\boldsymbol{S}_{i,j}^c = [\boldsymbol{s}_1^c, \ldots, \boldsymbol{s}_l^c, \boldsymbol{v}_i^a, \boldsymbol{v}_j^o]$, where $\boldsymbol{s}_{1:l}^a$, $\boldsymbol{s}_{1:l}^o$, and $\boldsymbol{s}_{1:l}^c$ are learnable pretix contexts initialized by "*a photo of*". Then these prompts are then fed into the frozen text encoder of CLIP [19] to obtain prompt features.

**Training Loss $\mathcal{L}^{\text{BAS}}$.** Following previous CZSL approaches [14, 15, 24], the parameters $\theta$ of the baseline model are learned by minimizing the three-path classification loss (Eq. 6) on the training dataset. Note that we have omitted the weight decay in Eq. 6 for simplicity. We just follow [14] set the weight decay as $5e{-}5$ for all our experiments.

**Feasibility Calibration for Open-World Setting.** Following [17, 14, 24], we adopt post-training feasibility calibration to filter out infeasible compositions that might be present in the open-world setting. The calibration relies on the assumption that similar objects tend to share similar attributes, while dissimilar objects are unlikely to exhibit shared attributes. Therefore, given a candidate pair $c = \langle a, o \rangle$, We calculate the feasibility compositions by computing the relationships between the objects and the attributes. First, we compute the similarities between the objects:

$$\rho_o(a, o) = \max_{\hat{o} \in \mathcal{O}^{se}} \frac{\phi(o) \cdot \phi(\hat{o})}{\|\phi(o)\| \|\phi(\hat{o})\|}, \tag{14}$$

where $\hat{o}$ is the other objects paired with the attribute $a$ in seen compositions, and $\phi(\cdot)$ is an embedding function that maps the primitive to a pre-trained embedding. We calculate the similarities $\rho_a(a, o)$ between attributes as same. Next, we combine the two similarities (*i.e.*, $\rho_o(a, o)$ and $\rho_a(a, o)$ ) with a pooling function to obtain $\rho(a, o)$.

Finally, we filter out infeasible compositions by only considering compositions above a threshold $\rho(a, o) > T$ on the validation set to make the prediction:

$$\hat{c} = \operatorname*{arg\,max}_{c_{i,j} \in \mathcal{C}^{tgt}, \rho(a_i, o_j) > T} p(c_{i,j}|x) + p(a_i|x) \cdot p(o_j|x). \tag{15}$$

# D  MORE QUANTITATIVE RESULTS

**Loss Coefficients $\alpha$ and $\beta$.** We further study the effect of loss coefficients $\alpha$ and $\beta$ for loss functions $\mathcal{L}_{\text{PCL}}$ (*cf.* Eq.10) and $\mathcal{L}_{\text{PDL}}$ (*cf.* Eq.11) on UT-Zappos [27]. In Table 6a, after fixing the loss coefficient $\beta$, CLUSPRO achieves the best performance when $\alpha$ is set to 0.2. Additionally, in Table 6b, we fix $\alpha$ and set $\beta$ with different values to test the impact of $\mathcal{L}_{\text{PDL}}$. We observe that setting $\beta$ as 0.5 leads to the best results across all metrics. Accordingly, we set $\alpha{=}0.2$ and $\beta{=}0.5$ in the training stage.

Table 6: The impact of loss coefficients $\alpha$ and $\beta$ (§D).

| Coefficient $\alpha$ | UT-Zappos | | | |
|---|---|---|---|---|
| | Seen↑ | Uneen↑ | HM↑ | AUC↑ |
| $\alpha = 0.1$ | 70.5 | 74.2 | 57.0 | 45.1 |
| $\alpha = 0.2$ | **70.7** | **76.0** | **58.5** | **46.6** |
| $\alpha = 0.3$ | 70.6 | 75.3 | 58.6 | 46.3 |
| $\alpha = 0.4$ | 70.4 | 75.2 | 58.2 | 46.2 |
| $\alpha = 0.5$ | 70.5 | 74.7 | 58.2 | 45.9 |

(a) Loss Coefficient $\alpha$

| Coefficient $\beta$ | UT-Zappos | | | |
|---|---|---|---|---|
| | Seen↑ | Uneen↑ | HM↑ | AUC↑ |
| $\beta = 0$ | 67.2 | 74.1 | 54.9 | 42.6 |
| $\beta = 0.5$ | **70.7** | **76.0** | **58.5** | **46.6** |
| $\beta = 1$ | 70.7 | 75.0 | 58.0 | 45.9 |
| $\beta = 5$ | 70.3 | 74.9 | 56.7 | 44.6 |
| $\beta = 10$ | 67.6 | 74.0 | 54.8 | 41.7 |

(b) Loss Coefficient $\beta$

**More Comparison Results with Existing CZSL Methods.** Apart from CLIP-based methods, we further compare our algorithm CLUSPRO with existing CZSL methods [29, 30, 7, 9, 32, 28, 20, 12, 31, 5] with a pre-trained ResNet18 [42] backbone across three datasets [26, 27, 28]. Table 7 and Table 8 report additional comparison results within *CW* and *OW* settings, respectively. As can be seen, CLIP-based methods significantly outperform traditional vision-based methods. This evidences that CLIP-based CZSL methods have stronger compositionality for zero-shot generalization. Notably, CLUSPRO surpasses all other methods and achieves state-of-the-art performance.

Table 7: **More comparison results**(§D) on MIT-States [26], UT-Zappos [27] and C-GQA [28] within *CW* setting.

| *Closed-World* | MIT-States | | | | UT-Zappos | | | | C-GQA | | | |
| Method | Seen↑ | Unseen↑ | HM↑ | AUC↑ | Seen↑ | Unseen↑ | HM↑ | AUC↑ | Seen↑ | Unseen↑ | HM↑ | AUC↑ |
| --- | --- | --- | --- | --- | --- | --- | --- | --- | --- | --- | --- | --- |
| *Traditional vision-based methods* | | | | | | | | | | | | |
| AoP [29] | 14.3 | 17.4 | 9.9 | 1.6 | 59.8 | 54.2 | 40.8 | 25.9 | 17.0 | 5.6 | 5.9 | 0.7 |
| LE+ [5] | 15.0 | 20.1 | 10.7 | 2.0 | 53.0 | 61.9 | 41.0 | 25.7 | 18.1 | 5.6 | 6.1 | 0.8 |
| TMN [30] | 20.2 | 20.1 | 13.0 | 2.9 | 58.7 | 60.0 | 45.0 | 29.3 | 23.1 | 6.5 | 7.5 | 1.1 |
| SymNet [7] | 24.2 | 25.2 | 16.1 | 3.0 | 49.8 | 57.4 | 40.4 | 23.4 | 26.8 | 10.3 | 11.0 | 2.1 |
| CompCos [9] | 25.3 | 24.6 | 16.4 | 4.5 | 59.8 | 62.5 | 43.1 | 28.1 | 28.1 | 11.2 | 12.4 | 2.6 |
| CGE [28] | 28.7 | 25.3 | 17.2 | 5.1 | 56.8 | 63.6 | 41.2 | 26.4 | 28.1 | 10.1 | 11.4 | 2.3 |
| Co-CGE [32] | 27.8 | 25.2 | 17.5 | 5.1 | 58.2 | 63.3 | 44.1 | 29.1 | 29.3 | 11.9 | 12.7 | 2.8 |
| SCEN [20] | 29.9 | 25.2 | 18.4 | 5.3 | 63.5 | 63.1 | 47.8 | 32.0 | 28.9 | 12.1 | 12.4 | 2.9 |
| CVGAE [31] | 28.5 | 25.5 | 18.2 | 5.3 | 65.0 | 62.4 | 49.8 | 34.6 | 28.2 | 11.9 | 13.9 | 2.8 |
| CANet [12] | 29.0 | 26.2 | 17.9 | 5.4 | 61.0 | 66.3 | 47.3 | 33.1 | 30.0 | 13.2 | 14.5 | 3.3 |
| CAPE [33] | 30.5 | 26.2 | 19.1 | 5.8 | 60.4 | 67.4 | 45.5 | 31.3 | 32.9 | 15.6 | 16.3 | 4.2 |
| *CLIP-based methods* | | | | | | | | | | | | |
| CLIP [19] | 30.2 | 46.0 | 26.1 | 11.0 | 15.8 | 49.1 | 15.6 | 5.0 | 7.5 | 25.0 | 8.6 | 1.4 |
| CoOp [82] | 34.4 | 47.6 | 29.8 | 13.5 | 52.1 | 49.3 | 34.6 | 18.8 | 20.5 | 26.8 | 17.1 | 4.4 |
| PCVL [39] | 48.5 | 47.2 | 35.3 | 18.3 | 64.4 | 64.0 | 46.1 | 32.2 | - | - | - | - |
| CSP [17] | 46.6 | 49.9 | 36.3 | 19.4 | 64.2 | 66.2 | 46.6 | 33.0 | 28.8 | 26.8 | 20.5 | 6.2 |
| DFSP(i2t) [18] | 47.4 | 52.4 | 37.2 | 20.7 | 64.2 | 66.4 | 45.1 | 32.1 | 35.6 | 29.3 | 24.3 | 8.7 |
| DFSP(BiF) [18] | 47.1 | 52.8 | 37.7 | 20.8 | 63.3 | 69.2 | 47.1 | 33.5 | 36.5 | 32.0 | 26.2 | 9.9 |
| DFSP(t2i) [18] | 46.9 | 52.0 | 37.3 | 20.6 | 66.7 | 71.7 | 47.2 | 36.0 | 38.2 | 32.0 | 27.1 | 10.5 |
| GIPCOL [83] | 48.5 | 49.6 | 36.6 | 19.9 | 65.0 | 68.5 | 48.8 | 36.2 | 31.9 | 28.4 | 22.5 | 7.1 |
| CDS-CZSL [15] | 50.3 | 52.9 | 39.2 | 22.4 | 63.9 | 74.8 | 52.7 | 39.5 | 38.3 | 34.2 | 28.1 | 11.1 |
| Troika [14] | 49.0 | 53.0 | 39.3 | 22.1 | 66.8 | 73.8 | 54.6 | 41.7 | 41.0 | 35.7 | 29.4 | 12.4 |
| PLID [16] | 49.7 | 52.4 | 39.0 | 22.1 | 67.3 | 68.8 | 52.4 | 38.7 | 38.8 | 33.0 | 27.9 | 11.0 |
| **CLUSPRO (Ours)** | **52.1**±0.6 | **54.0**±0.3 | **40.7**±0.2 | **23.8**±0.2 | **70.7**±1.0 | **76.0**±1.2 | **58.5**±0.6 | **46.6**±0.5 | **44.3**±0.2 | **37.8**±0.2 | **32.8**±0.2 | **14.9**±0.1 |

Table 8: **More comparison results**(§D) on MIT-States [26], UT-Zappos [27] and C-GQA [28] within *OW* setting.

| *Open-World* | MIT-States | | | | UT-Zappos | | | | C-GQA | | | |
| Method | Seen↑ | Unseen↑ | HM↑ | AUC↑ | Seen↑ | Unseen↑ | HM↑ | AUC↑ | Seen↑ | Unseen↑ | HM↑ | AUC↑ |
| --- | --- | --- | --- | --- | --- | --- | --- | --- | --- | --- | --- | --- |
| *Traditional vision-based methods* | | | | | | | | | | | | |
| AoP [29] | 16.6 | 5.7 | 4.7 | 0.7 | 50.9 | 34.2 | 29.4 | 13.7 | - | - | - | - |
| LE+ [5] | 14.2 | 2.5 | 2.7 | 0.3 | 60.4 | 36.5 | 30.5 | 16.3 | 19.2 | 0.7 | 1.0 | 0.1 |
| TMN [30] | 12.6 | 0.9 | 1.2 | 0.1 | 55.9 | 18.1 | 21.7 | 8.4 | - | - | - | - |
| SymNet [7] | 21.4 | 7.0 | 5.8 | 0.8 | 53.3 | 44.6 | 34.5 | 18.5 | 26.7 | 2.2 | 3.3 | 0.4 |
| CompCos [9] | 25.4 | 10.0 | 8.9 | 1.6 | 59.3 | 46.8 | 36.9 | 21.3 | 28.4 | 1.8 | 2.8 | 0.4 |
| CGE [28] | 29.6 | 4.0 | 4.9 | 0.7 | 58.8 | 46.5 | 38.0 | 21.5 | 28.3 | 1.3 | 2.2 | 0.3 |
| Co-CGE [32] | 26.4 | 10.4 | 10.1 | 2.0 | 60.1 | 44.3 | 38.1 | 21.3 | 28.7 | 1.6 | 2.6 | 0.4 |
| KG-SP [89] | 28.4 | 7.5 | 7.4 | 1.3 | 61.8 | 52.1 | 42.3 | 26.5 | 31.5 | 2.9 | 4.7 | 0.8 |
| CVGAE [31] | 27.3 | 9.9 | 10.0 | 1.8 | 58.6 | 48.4 | 41.7 | 22.2 | 26.6 | 2.9 | 6.4 | 0.7 |
| *CLIP-based methods* | | | | | | | | | | | | |
| CLIP [19] | 30.1 | 14.3 | 12.8 | 3.0 | 15.7 | 20.6 | 11.2 | 2.2 | 7.5 | 4.6 | 4.0 | 0.3 |
| CoOp [82] | 34.6 | 9.3 | 12.3 | 2.8 | 52.1 | 31.5 | 28.9 | 13.2 | 21.0 | 4.6 | 5.5 | 0.7 |
| PCVL [39] | 48.5 | 16.0 | 17.7 | 6.1 | 64.6 | 44.0 | 37.1 | 21.6 | - | - | - | - |
| CSP [17] | 46.3 | 15.7 | 17.4 | 5.7 | 64.1 | 44.1 | 38.9 | 22.7 | 28.7 | 5.2 | 6.9 | 1.2 |
| DFSP(i2t) [18] | 47.2 | 18.2 | 19.1 | 6.7 | 64.3 | 53.8 | 41.2 | 26.4 | 35.6 | 6.5 | 9.0 | 2.0 |
| DFSP(BiF) [18] | 47.1 | 18.1 | 19.2 | 6.7 | 63.5 | 57.2 | 42.7 | 27.6 | 36.4 | 7.6 | 10.6 | 2.4 |
| DFSP(t2i) [18] | 47.5 | 18.5 | 19.3 | 6.8 | 66.8 | 60.0 | 44.0 | 30.3 | 38.3 | 7.2 | 10.4 | 2.4 |
| GIPCOL [83] | 48.5 | 16.0 | 17.9 | 6.3 | 65.0 | 45.0 | 40.1 | 23.5 | 31.6 | 5.5 | 7.3 | 1.3 |
| CDS-CZSL [15] | 49.4 | 21.8 | 22.1 | 8.5 | 64.7 | 61.3 | 48.2 | 32.3 | 37.6 | 8.2 | 11.6 | 2.7 |
| Troika [14] | 48.8 | 18.7 | 20.1 | 7.2 | 66.4 | 61.2 | 47.8 | 33.0 | 40.8 | 7.9 | 10.9 | 2.7 |
| PLID [16] | 49.1 | 18.7 | 20.0 | 7.3 | 67.6 | 55.5 | 46.6 | 30.8 | 39.1 | 7.5 | 10.6 | 2.5 |
| **CLUSPRO (Ours)** | **51.2**±0.4 | **22.1**±0.2 | **23.0**±0.1 | **9.3**±0.2 | **71.0**±1.1 | **66.2**±1.0 | **54.1**±0.7 | **39.5**±0.8 | **41.6**±0.3 | **8.3**±0.2 | **11.6**±0.3 | **3.0**±0.1 |

**Evaluation Results for Models Pre-trained on Datasets with No Overlap.** To further highlight the robustness and superiority of our approach, we additionally present results under *CW* setting that utilize ViT-B backbone pre-trained with DINO [78] on ImageNet [90] in a self-supervised manner as ADE [23] instead of CLIP model [19]. Besides, we encode text representation with word2vec [91]

as [23, 12, 28] instead of the text encoder of CLIP. Table 9 reports the comparison results on UT-Zappos [27] and CGQA [28]. As seen, our algorithm also demonstrates better performance than the baseline and SOTA non-CLIP methods [20, 31, 33, 23].

Table 9: **More comparison results**(§D) on UT-Zappos [27] and C-GQA [28] within **CW** setting. Our algorithm utilizes ViT-B backbone pre-trained with DINO [78] as the visual encoder and word2vec [91] as the text encoder for a fair comparison with non-CLIP methods.

| Method | UT-Zappos | | | | C-GQA | | | |
|---|---|---|---|---|---|---|---|---|
| | Seen↑ | Unseen↑ | HM↑ | AUC↑ | Seen↑ | Unseen↑ | HM↑ | AUC↑ |
| AoP [29] | 59.8 | 54.2 | 40.8 | 25.9 | 17.0 | 5.6 | 5.9 | 0.7 |
| SCEN [20] | 63.5 | 63.1 | 47.8 | 32.0 | 28.9 | 12.1 | 12.4 | 2.9 |
| CVGAE [31] | 65.0 | 62.4 | 49.8 | 34.6 | 28.2 | 11.9 | 13.9 | 2.8 |
| CANet [12] | 61.0 | 66.3 | 47.3 | 33.1 | 30.0 | 13.2 | 14.5 | 3.3 |
| CAPE [33] | 60.4 | 67.4 | 45.5 | 31.3 | 32.9 | 15.6 | 16.3 | 4.2 |
| ADE [23] | 63.0 | 64.3 | 51.1 | 35.1 | 35.0 | 17.7 | 18.0 | 5.2 |
| CGE [28] | - | - | - | - | 38.0 | 17.1 | 18.5 | 5.4 |
| OADis [34] | - | - | - | - | 38.3 | 19.8 | 20.1 | 7.0 |
| Baseline | 61.0 | 62.9 | 45.1 | 31.9 | 34.6 | 15.9 | 16.6 | 4.5 |
| **CLUSPRO (Ours)** | **65.1** | **68.0** | **52.3** | **37.2** | **39.3** | **23.0** | **22.3** | **7.6** |

**Efficiency Analysis.** The efficiency comparison results with the state-of-the-art Trokia [14] and our baseline are reported in Table 10. Note that, CLUSPRO conducts within-primitive prototype clustering in a nonparametric manner and discards these learned sub-primitive prototypes during the testing phase. Thus, as shown in Table 10, CLUSPRO neither requires additional trainable parameters nor causes any inference delay during testing compared to the base model. Though efficient in terms of parameters and inference speed, our online clustering algorithm brings slight training delay ( ~ 11.5% on UT-Zappos [27]). Moreover, the effective clustering algorithm allows CLUSPRO to outperform the state-of-the-art Trokia in terms of classification accuracy, trainable parameters, and inference speed.

Table 10: Efficiency comparison on UT-Zappos [27]. Here, we report trainable parameters, training time per epoch, and inference speed for each model. See in §D for more details.

| Method | Params↓ | Memory↓ | Training time↓ | Inference Speed↓ | AUC↑ |
|---|---|---|---|---|---|
| Troika [14] | 21.7M | 19.9G | 4.1min | 14.9ms | 41.9 |
| Baseline | 8.7M | 18.2G | 4.0min | 14.6ms | 41.0 |
| **CLUSPRO (ours)** | 8.7M | 18.5G | 4.6min | 14.6ms | 46.6 |

**Number of Prototypes** $K$**.** In Table 11, we conduct the experiment by setting the number $K$ of prototypes based on the proportion of training samples for each primitive. In UT-zappos [27] dataset, training samples per primitive range from 0.2% to over 20%. Thus, we assign $K = 1$ to the primitive with $0.2 \sim 5\%$ training samples, $K = 2$ to the primitive with $5 \sim 10\%$ training samples, $K = 3$ to the primitive with $10 \sim 15\%$ training samples, $K = 4$ to the primitive with $15 \sim 20\%$ training samples, and $K = 5$ to the primitive with over 20% training samples. As seen, this approach results in slightly better performance than setting a fixed value for all the primitives.

Table 11: **Ablative experiments regarding varying** $K$ on UT-Zappos [27]. See §D for more details.

| $K$ range | UT-Zappos | | | |
|---|---|---|---|---|
| | Seen↑ | Uneen↑ | HM↑ | AUC↑ |
| unique value 5 | 70.7 | 76.0 | 58.5 | 46.6 |
| $[1, 5]$ | **71.0** | **76.0** | **58.6** | **46.8** |

# E   MORE QUALITATIVE VISUALIZATION

**More Case Study.** We provide additional success and failure cases of our method CLUSPRO across three CZSL benchmarks, *i.e.*, MIT-States [26] in Fig. 5, UT-Zappos [27] in Fig. 6 and C-GQA [28]

in Fig. 7. We also compare our approach CLUSPRO with baseline without within-primitive clustering. As seen, by mining rich sub-primitive patterns via within-primitive clustering, CLUSPRO can produce more accurate composition predictions, even recognizing fine-grained primitives, such as various materials and colors. For failure cases, where the attribute and object of images are highly entangled, CLUSPRO still identifies the part of the attribute-object composition.

## F    PSEUDO CODE OF PROTOTYPE ASSIGNMENT AND UPDATING

Algorithm 1 provides the pseudo-code of "Local-aware Prototype Assignment" and "Prototype Updating". To guarantee reproducibility, our code is available at CLUSPRO.

## G    DISCUSSION

**Data Overlap Analysis.** Given that CLIP [19] is trained on millions of text-image pairs sourced from the web, it is hard to know whether CLIP has been exposed to certain unseen compositions during its pre-training, which violates the zero-shot learning setting factually. Most current researches [18, 17, 15, 24] in CZSL, including our work, report the performance in the Generalized Zero-shot Learning [9] for both *CW* and *OW* settings, where test samples include both seen and unseen compositions. Hence, it naturally brings up the question: *whether CLIP meets the definition of Generalized Zero-shot Learning*. Based on the data overlap analysis on 35 datasets as reported in [19], there is a median overlap of 2.2% and an average overlap of 3.2%. Due to this small amount of overlap, the overall accuracy shift is less than 0.1% with the largest shift as 0.6%. As such, CLIP is only exposed to a very small number of unseen compositions during pre-training, and the impact on the performance is limited. However, the potential composition leaking in the pre-training of CLIP indeed leads to an unfair comparison with other non-CLIP methods [23, 28, 34]. Thus, we argue that it is important to emphasize the comparisons with other CLIP-based methods that share the same pre-training (comparison results in Table 1 and 2). Moreover, where possible, it is also advisable to report performance metrics for non-CLIP variants to ensure a comprehensive evaluation.

**Limitation.** One limitation of our algorithm is that it needs extra within-primitive prototype clustering from the perspective of optimal transport after each training iteration, leading to increasing time complexity. However, in practice, our clustering algorithm only brings slight training delay attributed to efficient GCG algorithm [73] for solving such clustering problem. Additionally, our mined sub-primitive prototypes are subject to the data distribution of the training dataset. Thus rare primitive concepts in the dataset (*i.e.*, long-tail distribution), like many previous state-of-the-arts, pose significant challenges for primitive-wise clustering to discover diverse sub-primitive patterns, thus resulting in poor performance on unseen compositions about these primitives. Also, the number of prototypes for each primitive currently is set to a fixed value, which may not be optimal given that intra-primitive variability varies across primitives. Thus it is interesting to find ways to automatically determine $K$ [92] for different primitives, which may further boost performance.

**Border Impact.** This work introduces CLUSPRO, a powerful clustering-based framework for Compositional Zero-Shot Learning via exploring dataset-level context, which overcomes the limitations of previous solutions relying on single or paired images for visual disentanglement. This model provides a feasible way to discover diverse sub-primitive patterns in massive training data, and directly shape well-structured embedding space based on these mined patterns. On the positive side, CLUSPRO pushes the boundary of CZSL algorithms, and can benefit a number of potential real-world applications, *e.g.*, autonomous driving and robotics. For the potential negative societal impacts, our CLUSPRO struggles in handling very rare primitives in the dataset, which is a common issue of current CZSL algorithms, thus leading to inaccurate decisions or planning of systems. To avoid this potential problem, it is crucial to develop a security protocol in case our approach fails to perform as expected in real-world scenarios.

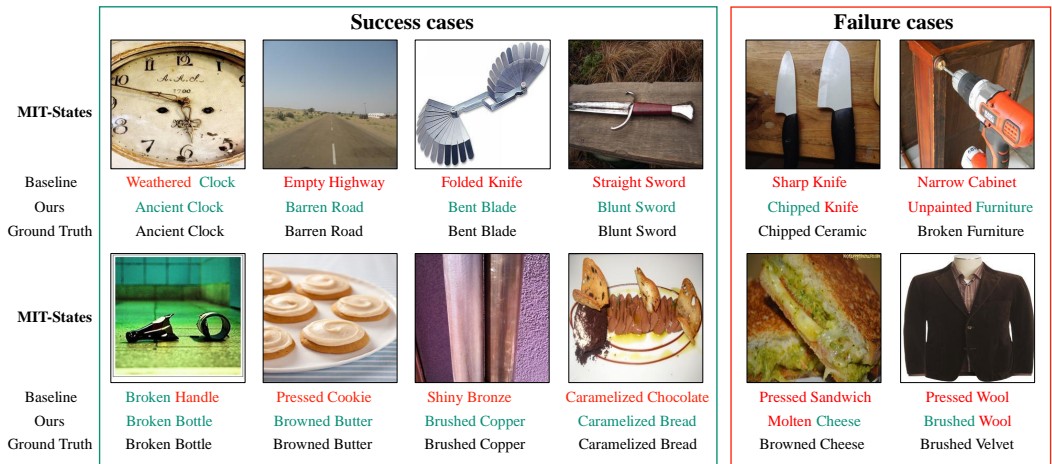

Figure 5: More case studies on MIT-States [26]. We compare CLUSPRO with baseline without primitive-wise prototype clustering. Correct and incorrect predictions are marked in **green** and **red**, respectively.

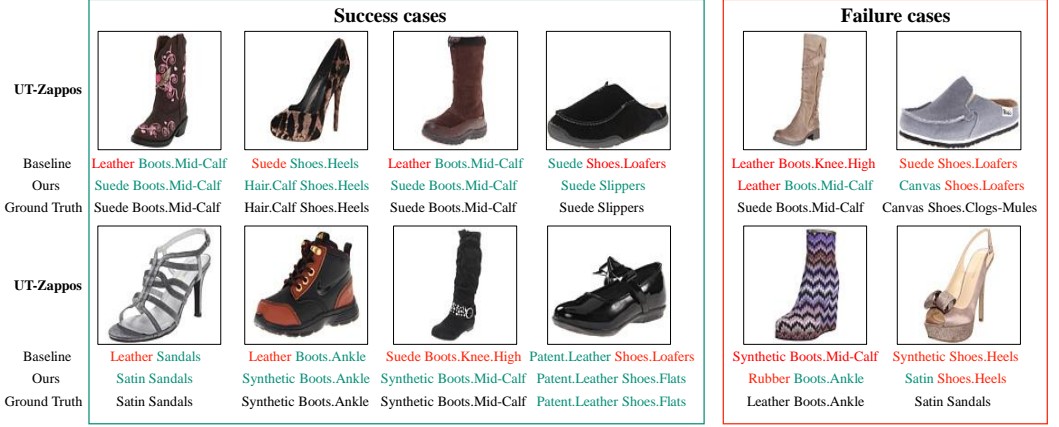

Figure 6: More case studies on UT-Zappos [27]. We compare CLUSPRO with baseline without primitive-wise prototype clustering. Correct and incorrect predictions are marked in **green** and **red**, respectively.

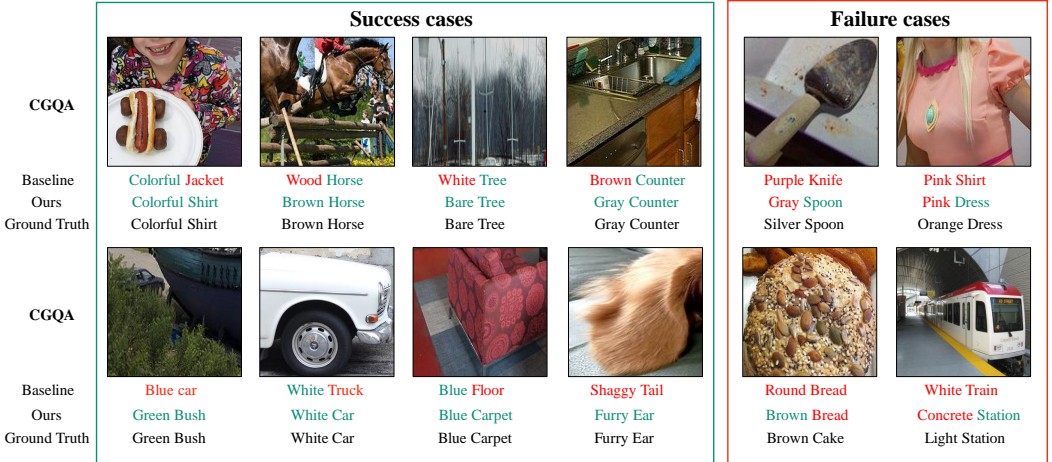

Figure 7: More case studies on C-GQA [28]. We compare CLUSPRO with baseline without primitive-wise prototype clustering. Correct and incorrect predictions are marked in **green** and **red**, respectively.

---

**Algorithm 1** Pseudo-code of prototype assignment and updating in a PyTorch-like style.

---

```
"""
# P: primitive prototypes (C x K x D)
# F: primitive feature embeddings (N x D)
# y: labels for primitive features F (N)

# C: number of attribute or object primitives
# K: number of prototypes for each primitive
# N: batch size
# mu: momentum coefficient (Eq.8)
"""

#======= local-aware prototype assignment =======#
def prototype_assignment(F, label):
    # prototype assignment for each primitive feature (Eq.7)
    Q = torch.einsum('nd,ckd->nkc', F, P)
    S = torch.einsum('nd,nd->nn', F, F) # primitive self-similarity matrix

    for c in range(C):
        init_q = Q[...,c]
        init_l = local_online_clustering(init_q, S) # one-hot matrix

        # prototype assignments for features in primitive c
        l_c = Q[label == c]
        f_c = F[label == c, ...]

        # find features that are assigned to each sub-primitive prototype
        l_c_tile = torch.einsum(l_c, tile=K)
        l_q = init_l * l_c_tile

        # find features with primitive c that are correctly classified
        f_c_tile = repeat(l_c, tile=f_c.shape[-1])
        f_c_q = f_c * f_c_tile

        # new cluster features for primitive c
        a = torch.mm(l_q.transpose(),f_c_q)

        # momentum updating for each primitive prototype (Eq.8)
        prototype_updating(l_q, a, c)

#======= prototype updating =======#
def prototype_updating(l_q, a, c):
    # num assignments for each prototype of primitive c
    n = torch.sum(l_q, dim=0)
    a = normalize(a)

    # prototype updating
    if torch.sum(n) > 0:
        P_c = P[c, n != 0,:] * mu + a[n != 0,:] * (1 - mu)
        P[c, n != 0, :] = P_c
```

---

