# OpenReview forum: "Learning Clustering-based Prototypes for Compositional Zero-Shot Learning"
_ICLR.cc/2025/Conference — ICLR 2025 Poster_

### Official Review · Reviewer_Q4uv · 2024-11-01

**Soundness:** 2
**Presentation:** 2
**Contribution:** 2
**Rating:** 6
**Confidence:** 4

**Summary:**

This paper proposes a clustering-based prototype mining framework for CZSL which conducts within-primitive clustering on the embedding space for automatically discovering and dynamically updating prototypes. These representative prototypes are subsequently used to repaint a well-structured and independent primitive embedding space, ensuring intra-primitive separation and inter-primitive decorrelation through prototype-based contrastive learning and decorrelation learning. Experiments on three benchmarks demonstrate CLUSPRO outperforms top-leading CZSL solutions under both closed-world and open-world settings.

**Strengths:**

- Overall this paper is well-written and is easy to understand.
- The motivation is reasonable and the proposed framework is a little novel for CZSL.
- The CLUSPRO outperforms top-leading CZSL solutions under both closed-world and open-world settings on three benchmarks.

**Weaknesses:**

- There is a lack of discussion and citation of some related works. In ZSL/CZSL fields, prototype learning is wide adopted in the works [A][B]. The Local-aware Prototype Assignment module is also similar to the previous work [C], but the necessary discussion is absent.

- The overall novelty is limited. The Local-aware Prototype Assignment technique is adopted in the work [C], and the Prototype-based Contrastive Learning have been used in previouse works (e.g., [A]).

- Writing of Chapter 3.3 could be improved. The authors do not make it clear how the K prototypes are obtained. With reference to the context, is it means k-means clustering results on visual fetaures?  In addition, when the variable notations appear for the first time, the authors should give an obvious comment (e.g., $n$ in $f_{n}$) for reducing ambiguity.

I will consider updating the score if the authors can address my concerns well.

[A] ProtoCLIP: Prototypical Contrastive Language Image Pretraining

[B] Dual progressive prototype network for generalized zero-shot learning

[C] Rethinking semantic segmentation: A prototype view

**Questions:**

In the Prototype-based Contrastive Learning, the strategy encourages each primitive feature $f_n$ to
be similar to its assigned prototype $P+$ and dissimilar to all other $K(M+N)−1$ irrelevant prototypes
$P-$. It means that the $K-1$ prototypes belonging to the same primitive but with different assignments are treated as negative samples in the same way as prototypes  $K(M+N-1)$ of other primitives.
Does this have a  impact on representation learning?

---

> ### Author Response · Authors · 2024-11-22
> **Point-to-Point Response to Reviewer Q4uv (1/2)**
>
> We thank reviewer Q4uv for the valuable time and constructive feedback. We provide point-to-point response below.
>
>
> ---
>
>
>
>
>
> **Q1: Lack of discussion and citation of some related works.**
>
>
>
> **A1**: Thank you for pointing this out! To better address your concern and avoid potential misunderstandings, we expand and further discuss the related work [A, B, ref1, ref2] on prototype learning for ZSL/CZSL fields in L131-134 and the difference between our local-aware prototype assignment module and previous work [C, ref3] in L251-255 to enhance the contextual relevance and depth of our review, as follows:
>
> > **L131-134**: In the (compositional) zero-shot learning field, these works [A, B, ref1, ref2] extensively explore prototype learning to enhance feature representation. However, they typically model each class with only one prototype and their prototypes are often flexible parameters, ignoring intra-class diversity.
>
>   >**L251-255**: Different from the classical formulation in [C, ref3], *i.e.*, Optimal Transport with entropic constraints, our local-aware prototype assignment can produce superior assignments by fully considering the intrinsic coherence structure of attribute feature distribution, *i.e.*, intra-distribution coherence.
>
>
>
>
> [A] ProtoCLIP: Prototypical Contrastive Language Image Pretraining. TNNLS 2023.
>
>
> [B] Dual progressive prototype network for generalized zero-shot learning. NeurIPS 2021.
>
> [C] Rethinking semantic segmentation: A prototype view. CVPR 2022.
>
>
> [ref1] Visual-augmented dynamic semantic prototype for generative zero-shot learning. CVPR 2024.
>
> [ref2] Attribute prototype network for zero-shot learning. NeurIPS 2020.
>
>
>
> [ref3] Prompt learning with optimal transport for vision-language model. arXiv 2022.
>
>
>
>
>
> ---
>
>
>
>
>
> **Q2: Novelty of ClusPro.**
>
>
>
> **A2**: Sorry for this confusion.
>
> **First**, although ClusPro and [C] both reformulate the prototype assignment problem as an optimal transport problem, as proposed by [ref4], there are notable differences in their implementations.
>
> [C] utilizes the classical formulation, *i.e.*, Optimal Transport with entropic constraints, defined as $\Omega({L}^{a\top})=-\sum l^a_{n,k}\log(l^a_{n,k})$, which typically yields denser transportation plans. The entropic regularization term enables [C] to approximate the optimal transport problem by the Sinkhorn algorithm.
>
>
> Conversely, ClusPro adopts **another formulation with stronger priors**, *i.e.*, Optimal Transport with a structure-aware constraint, defined as $\Omega({L}^{a\top})=-\langle{S}^{a},({L}^{a}\odot{Q}^{a}{)^\top}{({L}^{a}\odot{Q}^{a})} \rangle$, which promotes intra-distribution coherence by encouraging local consistency (see L254-257). Given that this objective function is non-convex, a **Generalized Conditional Gradient (GCG) algorithm** [ref5] is utilized to effectively solve the problem. Moreover, as illustrated in Table 4d (L446-450), it can be observed that our local-aware prototype assignment demonstrates superior performance compared to classical Optimal Transport [C].
>
>
> **Second**, [A] only employs **instance-level** contrastive learning, which aligns with the standard CLIP framework. For prototype, it adopts K-means clustering akin to DeepCluster, where cluster assignments serve as pseudo-labels, and a **cross-entropy objective** is optimized to learn robust representations.
>
>
>
> In contrast, ClusPro implements **prototype-level** contrastive learning (PCL) to shape well-structured embedding space. It optimizes an **infoNCE objective** to maximize the prototype assignment posterior probability.
>
>
>
> As outlined in L303-311, our prototype-based contrastive learning offers two key advantages over previous methods like [A]:
>
> 1. It constructs positive and negative pairs using clustering-based prototypes, thereby avoiding the need for sophisticated negative sampling strategies.
>
> 2. It leverages pre-constructed prototypes for contrast computation, thus avoiding additional computational and storage costs.
>
>  Moreover, *Prototype-anchored Decorrelation Learning* is also devised to complement prototype-based contrastive learning, working together to promote intra-primitive separation and inter-primitive decorrelation.
>
> In summary, motivated by the objective to "model each primitive with a set of prototypes", we propose a framework for CZSL that includes prototype construction and prototype-based self-supervised learning to achieve intra-primitive separation and inter-primitive correlation in the embedding space. Thus, our approach stands out from previous works with clear distinctions in both the **core motivation** (acknowledged by the reviewers siAz and 1rZT) and **techniques** (acknowledged by the reviewer siAz) employed.
>
> [ref4] Self-labelling via simultaneous clustering and representation learning. ICLR 2020.
>
>
> [ref5] Generalized conditional gradient: analysis of convergence and applications. arXiv 2015.

---

> ### Author Response · Authors · 2024-11-22
> **Point-to-Point Response to Reviewer Q4uv (2/2)**
>
> **Q3: Writing of Sec. 3.3 could be improved.**
>
> Our apologies. We make detailed clarification below and revise Section 3.3 to improve clarity.
>
> ---
>
>
>
> **Q3.1: Clarify how the *K* prototypes are obtained.**
>
>
>
> **A3.1**: Sorry for this confusion. In fact, we do not obtain the prototypes through the off-the-shelf K-means algorithm [ref6], which is based on data reconstruction; instead, we **cast the prototype assignments as an optimal transport problem**, following the learning formulation proposed by [ref7]. Compared to methods such as K-means, the new formulation [ref7] is more principled and allows to more easily demonstrate properties of the method such as convergence.
>
>
>
> As detailed in L235-268, given attribute features ${F}^{a} \in  \mathbb{R}^{D\times N^{a}}$, our goal is to seek the optimal assignment matrix ${L}^{a} \in  \mathbb{R}^{K\times N^{a}}$, thereby deriving the prototypes ${P}^{a}={F}^{a}{{L}^{a}}^\top  \in  \mathbb{R}^{D\times K}$. The clustering within each attribute can be achieved by the optimization of the assignment matrix ${L}^{a}$:
>
>
>
> $\min\_{{L}^{a}\in  \mathcal{L}^{a}}\langle{L}^{a\top},-\log{Q}^{a}\rangle + \kappa\Omega({L}^{a\top})$.
>
>
>
> The relevant contents in Sec. 3.3 in L258-260 have been modified to reduce confusion and improve readability. Thanks.
> > **L258-260**:  Unlike offline clustering [73, 74] requiring multiple passes over the entire dataset for feature computation, we cast local-aware prototype assignment as an optimal transport problem, so as to scale our algorithm to massive data by online clustering.
>
>
> [ref6] Deep clustering for unsupervised learning of visual features. ECCV 2018.
>
> [ref7] Self-labelling via simultaneous clustering and representation learning. ICLR 2020.
>
> ---
>
>
>
>
>
> **Q3.2: Clarify the variable notations.**
>
>
>
> **A3.2**: Thank you so much for your careful review! We make an obvious comment in the revised version for the variable notations to improve clarity, *e.g.*, ${p}^{a}\_k$, ${f}^{a}\_n$ and ${f}\_n$.
>
>   1. ${p}^{a}\_k$: "${p}^{a}\_k$ is $k$-th prototypes of attribute $a$"
>   2. ${f}^{a}\_n$: "${f}^{a}\_n$ is $n$-th attribute features of attribute $a$"
>   3. ${f}\_n$: "${f}\_n$ is $n$-th primitive features"
>
>
>
>
> ---
>
>
>
> **Q4: Use **$K(M+N-1)$ irrelevant prototypes** from other primitives as negative samples.**
>
>
>
> **A4**: Thanks for your thoughtful comments on our manuscript. As outlined in L293-296, our prototype-based contrastive learning is designed to capture intra-primitive separation. To this end, it is pivotal to ensure the discriminativeness of prototypes within each primitive, *i.e.*, to enforce each feature to be similar with its assigned prototype $p_+$ and dissimilar with other $K-1$ irrelevant prototypes. To increase the scale of negative samples, we add $K(M+N-1)$ irrelevant prototypes from other primitives, ultimately resulting in a total of $K(M+N)-1$ irrelevant prototypes $\mathcal{P}_{-}$. Additionally, using only $K(M+N-1)$ irrelevant prototypes from other primitives as negative samples can ensure inter-primitive separation to some extent, but it does not guarantee intra-primitive separation. We incorporate the discussions into Sec. 3.4. Thanks.
>
>
>
>
>
> ---
>
> Finally, we would like to sincerely thank you for your careful review and valuable comments, which really help us to improve this work. We hope we addressed your concerns. Please let us know if you'd like any further information.

---

> > ### Comment · Reviewer_Q4uv · 2024-11-25
> >
> > Thanks for the detailed response. Overall, my concerns are addressed. If possible, I expect the author to provide code for results reproduction. I will consider to update the score before the ddl.

---

### Official Review · Reviewer_Aocv · 2024-11-02

**Soundness:** 2
**Presentation:** 2
**Contribution:** 2
**Rating:** 6
**Confidence:** 5

**Summary:**

This paper proposes clustering-based method for compositional zero-shot learning.
Specifically,

**Strengths:**

Clear presentation

**Weaknesses:**

Major problem:

Problem1. The definition of Compositional Zero-shot Learning (CZSL) is not correctly used.
The authors mentioned CZSL requires the model to recognize unseen compositions without additional training data, but in your method, you use pretrained CLIP trained by large-scale dataset: CLIP is trained from millions text-image pairs from web.
In other word, all your claimed unseen compositions are seen by CLIP actually.
Thus, a key question is emerged: if your visual&text encoders never meet your test compositions, can the seen and unseen compositions be seperated?
Please provide the results that using other visual&text encoders or using vision-language models whose pretrained training datasets have no overlap with the evaluted datasets.

Problem2. Besides, in the original paper of CLIP, it has clearly claimed they changed the definition of ZSL from class-level to dataset-level. "In computer vision, zero-shot learning usually refers to the study of generalizing to unseen object categories in image classification (Lampert et al., 2009). We instead use the term in a broader sense and study generalization to unseen datasets." ﻿ This excerpt is from Section 3.2 of the CLIP paper. There is a significant gap between CLIP’s definition of zero-shot learning and the CZSL task you are addressing, which still focuses on classifying unseen composition categories. This difference is substantial and cannot be overlooked.

In a short, I suggest the proposed method is a transductive classification method insteat of ZSL method actually since the pre-trained CLIP have been exposed to numerous unseen composition samples.

Problem3.
You only compare your method to CLIP-based CZSL methods, but there are many methods that do not use CLIP.
I know those non-CLIP methods perform worsely than CLIP-based methods.
But, as I said in Problem1&2, since CLIP have been exposed to numerous unseen composition samples, the experiments are extramely unfair for other non-CLIP SOTAs.

Problem4. Why only consider the diversities of the attribute, but do not for the objects?

**Questions:**

See weakness

**Details Of Ethics Concerns:**

The paper uses a large-scale pretrained visual-language model CLIP. Thus, the bias problem must be considered.

---

> ### Author Response · Authors · 2024-11-22
> **Point-to-Point Response to Reviewer Aocv (1/2)**
>
> We thank reviewer Aocv for the valuable time and constructive feedback. We provide point-to-point response below.
>
> ---
>
>
> **Q1.1: Clarify the definition of Compositional Zero-shot Learning (CZSL).**
>
>
>
>
>
> **A1.1**: Sorry for this confusion. Actually, most current researches [ref1, ref2, ref3, ref4, ref5] in compositional zero-shot learning (CZSL), including our work (see L150-157), report the performance in the **Generalized Zero-shot Learning** [ref6] for both the closed-world and the open-world settings, where **test samples include both seen and unseen compositions**.
>
> Given that CLIP is trained on millions of text-image pairs sourced from the web, it is hard to know whether CLIP has been exposed to certain unseen compositions during its training. However, we noted that the CLIP paper performed a data overlap analysis (Sec. 5 of [ref7]) w.r.t. 35 datasets on zero-shot transfer experiments. Finally, they find there is a median overlap of 2.2% and an **average overlap of 3.2%**. According to [ref7], if overlap exists, **overall accuracy shift is less than 0.1% with the largest shift as 0.6%**.
>
> In light of above discussion, we argue that CLIP-based model still **aligns with the definition of CZSL**, even though CLIP might may be exposed to a small number of unseen compositions during pre-training. Besides, given the prevalence of CLIP-based approaches in the CZSL, it seems a little unfair that treating this as a specific limitation of our approach.
>
> Moreover, the experiment results in Table 7 and 8 shows that CLIP without any fine-tuning underperforms task-specific architectures (*i.e.*, non-CLIP methods) in the CZSL task, even though it has been pre-trained on vastly more data.  We suspect that this is because **contrastive pre-training on Web image captions does not require the model to learn precise compositional semantics**. Thus, it is non-trivial to explore how to explicitly enforce compositional semantics by fine-tuning CLIP.
>
> [ref1] Decomposed soft prompt guided fusion enhancing for compositional zero-shot learning. CVPR 2023.
>
> [ref2] Learning to compose soft prompts for compositional zero-shot learning. ICLR 2023.
>
> [ref3] Context-based and Diversity-driven Specificity in Compositional Zero-Shot Learning. CVPR 2024.
>
> [ref4] Prompting language-informed distribution for compositional zero-shot learning. ECCV 2024.
>
> [ref5] Retrieval-Augmented Primitive Representations for Compositional Zero-Shot Learning. AAAI 2024.
>
> [ref6] Open world compositional zero-shot learning. CVPR 2021.
>
> [ref7] Learning Transferable Visual Models From Natural Language Supervision. ICML 2021.
>
> ---
>
>
>
> **Q1.2: Evaluation results for models pre-trained on datasets with no overlap.**
>
>
>
> **A1.2**: To further highlight the robustness and superiority of our approach, we additionally present results on CW setting that **utilizing ViT-B backbone pre-trained with DINO [ref8] on ImageNet in a self-supervised manner** as ADE [ref9] instead of CLIP model, as follows:
>
> |Method|UT-Zappos||||CGQA||||
> |:-|:-|:-|:-|:-|:-|:-|:-|:-|
> ||Seen|Unseen|HM|AUC|Seen|Unseen|HM|AUC|
> |CANet | 61.0| 66.3| 47.3| 33.1 |30.0 |13.2| 14.5| 3.3|
> |CAPE | 60.4| 67.4 |45.5 |31.3| 32.9| 15.6| 16.3 |4.2|
> |ADE|63.0|64.3|51.1|35.1|35.0|17.7|18.0|5.2|
> |CGE| -| - |- |-| 38.0 |17.1|18.5|5.4 |
> |OADis|-| - |- |-| 38.3| 19.8|20.1|7.0 |
> |Baseline|61.0|62.9|45.1|31.9|34.6|15.9|16.6|4.5|
> |Ours|$\mathbf{65.1}$ |$\mathbf{68.0}$ |$\mathbf{52.3}$ |$\mathbf{37.2}$ |$\mathbf{39.3}$ |$\mathbf{23.0}$ |$\mathbf{22.3}$ |$\mathbf{7.6}$ |
>
>
> As seen, **our model also demonstrates better performance than the baseline and SOTA non-CLIP methods (e.g.,  CANet, CGE, OADis and  ADE ),when using the backbone pre-trained on datasets with no overlap**. We incorporate the above results into the Appendix Sec.D and Table 9. Thanks.
>
>
> [ref8] Emerging properties in self-supervised vision transformers. ICCV 2021.
>
> [ref9] Learning Attention as Disentangler for Compositional Zero-shot Learning. CVPR 2023.
>
> ---
>
>
>
> **Q2: The usage of CLIP contradicts the definition of CZSL.**
>
>
>
> **A2**: Thanks for your careful review. Please see **A1.1** for detailed clarification. Thus, we safely conclude that CLIP-based methods still align with the definition of CZSL. Thanks.

---

> ### Author Response · Authors · 2024-11-22
> **Point-to-Point Response to Reviewer Aocv (2/2)**
>
> **Q3: Non-CLIP version of ClusPro vs. Non-CLIP methods.**
>
>
>
> **A3**: Thanks for your comment. We would like to respectfully clarify that we had provided comparison results with non-CLIP methods in Table 7 and 8. It is worth noting that some CLIP-based methods (*i.e.*, coop, CSP) perform worse than non-CLIP methods under certain settings. For example, CSP performs worse than non-CLIP method kG-SP on UT-zappos under OW setting.
>
>
>
> In addition, for the sake of fair comparison with non-CLIP SOTAs, we also present results that do not utilize CLIP model. Please see **A1.2** for experimental results. The results are added in Appendix Sec.D and Table 9. Thanks.
>
>
>
> ---
>
>
>
> **Q4: The diversities of primitive (i.e., attribute and object).**
>
>
>
> **A4**: Sorry for this confusion. In fact, we consider **both the diversity of attributes and objects**:
>
>
>
> - As stated in L228-230: Notably, we present the online primitive-wise clustering process **within both the attribute and object embedding spaces**. For clarity, we only explain the prototype construction in the attribute branch, while the object branch follows the same process.
>
>
> - As detailed in L282-283: we **construct a set of prototypes for each attribute, and each object**, to represent diverse sub-primitive patterns.
>
>
> But, for simplicity, we primarily highlight attribute diversity in our exposition. To improve clarity, the following updates have been made to the manuscript:
>
> > **L228-230**: Notably, we present the online primitive-wise clustering process within both the attribute and object embedding spaces, so as to well represent rich and diverse patterns within each primitive.
>
>
>
> ---
>
> Finally, we would like to sincerely thank you for your careful review and valuable comments. We have revised our manuscript with discussions and experimental results according to your review. It is welcome if there are any further questions or suggestions.

---

> ### Comment · Reviewer_Aocv · 2024-11-24
> **Official Comment by Reviewer Aocv**
>
> I appreciate the response from authors.
> It addresses my partial concerns.
> **If authors can resolve the remained issues, I am willing to increase my score**.
>
> 1. Although the visual encoder is replaced, the text encoder still is from CLIP. It still might bring potential risk for composition leaking.
> Could you provide the results that use other text encoder, even just on one dataset, before the dicussion close?
>
> 2. Although CLIP provides a data overlap analysis, but it violated the zero-shot learning setting factually.
> To prevent future researchers from having similar issues, could you provide a highlighted brief claim (e.g. claim in footnote) about the overlap discussion and link the detailed discussion and results in appendix？

---

> > ### Author Response · Authors · 2024-11-25
> > **Response to Reviewer Aocv**
> >
> > We greatly appreciate your timely feedback. Below is our response to your new comments.
> >
> > ---
> >
> > **Q1: Clarify the implementation of text encoder.**
> >
> > **A1**:  Sorry for this confusion. Actually, the text encoder of ClusPro in Table 9 is implemented by **word2vec [ref1]** instead of the text encoder of CLIP, following [ref2, ref3, ref4]. To improve clarity, we supplement the implementation of text encoder used into Appendix Sec. D and Table 9, as follows:
> >
> > >**L971-972** Besides, we encode text representation with word2vec [ref1] as [ref2, ref3, ref4] instead of the text encoder of CLIP.
> >
> > >**L976-978**  Our algorithm utilizes ViT-B backbone pre-trained with DINO as the visual encoder and word2vec [ref1] as the text encoder for a fair comparison with non-CLIP methods.
> >
> > [ref1] Distributed representations of words and phrases and their compositionality. NeurIPS 2013.
> >
> > [ref2] Learning attention as disentangler for compositional zero-shot learning. CVPR 2023.
> >
> > [ref3] Learning conditional attributes for compositional zero-shot learning. CVPR 2023.
> >
> > [ref4] Learning graph embeddings for compositional zero-shot learning. CVPR 2021.
> >
> > ---
> >
> > **Q2: Data overlap analysis about CLIP.**
> >
> > **A2**: Certainly, we appreciate your insightful feedback. In the footnote on page 1, we make a brief claim regarding the overlap discussion, which refers the reader to a more detailed discussion in the appendix Sec. G.
> >
> > >**Footnote on Page 1**: Given that CLIP might be exposed to certain unseen compositions during pre-training, we provide detailed data overlap discussion in the appendix Sec. G.
> >
> > >**Appendix Sec. G**: **Data Overlap Analysis**. Given that CLIP [19] is trained on millions of text-image pairs sourced from the web, it is hard to know whether CLIP has been exposed to certain unseen compositions during its pre-training, which violates the zero-shot learning setting factually. Most current researches [18, 17, 15, 24] in CZSL, including our work, report the performance in the Generalized Zero-shot Learning [9] for both *CW* and *OW* settings, where test samples include both seen and unseen compositions. Hence, it naturally brings up the question: *whether CLIP meets the definition of Generalized Zero-shot Learning*. Based on the data overlap analysis on 35 datasets as reported in [19], there is a median overlap of 2.2% and an average overlap of 3.2%. Due to this small amount of overlap, the overall accuracy shift is less than 0.1% with the largest shift as 0.6%.
> > As such, CLIP is only exposed to a very small number of unseen compositions during pre-training, and the impact over the performance is limited. However, the potential composition leaking in the pre-training of CLIP indeed leads to an unfair comparison with other non-CLIP methods [23, 28, 34]. Thus, we argue that it is important to emphasize the comparisons with other CLIP-based methods that share the same pre-training (comparison results in Table 1 and 2). Moreover, where possible, it is also advisable to report performance metrics for non-CLIP variants to ensure a comprehensive evaluation.
> >
> >
> >
> > ---
> >
> > We are grateful for your thorough review of our submission and the insightful feedback you've provided. Please let us know if you'd like any further information.

---

> > > ### Comment · Reviewer_Aocv · 2024-11-25
> > > **Official Comment by Reviewer Aocv**
> > >
> > > The authors addressed all my considerations. I changed my rating "5->6".

---

> > > > ### Author Response · Authors · 2024-11-26
> > > > **Response to Reviewer Aocv**
> > > >
> > > > Dear Reviewer Aocv,
> > > >
> > > >
> > > > Thank you again for your positive feedback and for considering our rebuttal. Please let us know if you'd like any further information.
> > > >
> > > > Sincerely yours,
> > > >
> > > > Authors.

---

### Official Review · Reviewer_1rZT · 2024-11-03

**Soundness:** 3
**Presentation:** 3
**Contribution:** 2
**Rating:** 6
**Confidence:** 3

**Summary:**

This paper reveals that exsiting compositional zero-shot learning (CZSL) methods only consider an isolated centroid for each primitive, ignoring rich and diverse intra-primitive patterns. To address this, the authors propose a clustering-based method CLUSPRO to learn a well-structured and independent embedding space with multiple discriminative prototypes for each primitive, thus improving CZSL. Specifically, CLUSPRO alternates between two steps: 1) performing within-primitive online clustering to automatically discover and dynamically update prototypes, and 2) using prototype-based primitive representation learning to encourage intraprimitive separation and inter-primitive decorrelation.

**Strengths:**

- This paper is well-motivated, highlighting that a single centroid primitive representation demonstrates limited tolerance to intraprimitive variance.
- The proposed method CLUSPRO is reasonable and has a good performance across three benchmarks under both closed-world and open-world settings.
- The authors present a comprehensive analysis of their method CLUSPRO.

**Weaknesses:**

- My major concern is the number of prototype. In the experiments, the number of prototypes $K$ is empirically set to 5, which may not ne a good strategy. Additionally, using the same number of prototypes for all attributes seems unreasonable. Are there any automatic methods for discovering prototypes, thus mitigating the need for manually setting this number?
- Does performing clustering on the entire dataset will face resource limitations when dealing with very large datasets?

**Questions:**

Please see the weaknesses.

---

> ### Author Response · Authors · 2024-11-22
> **Point-to-Point Response to Reviewer 1rZT**
>
> We thank reviewer 1rZT for the valuable time and constructive feedback. We provide point-to-point response below.
>
> ---
>
> **Q1: Number of prototypes for each primitive.**
>
>
> **A1**: Thanks for pointing out this promising direction. Intra-primitive variability indeed varies across primitive. Considering this, we conduct the experiment by setting the number $K$ of prototypes based on the proportion of training samples for each primitive. In UT-Zappos dataset, training samples per primitive range from 0.2\% to over 20\%. Thus, we assign $K=1$ to the primitive with 0.2-5\% training samples, $K=2$ to the primitive with 5-10\% training samples, $K=3$ to the primitive with 10-15\% training samples, $K=4$ to the primitive with 15-20\% training samples, and $K=5$ to the primitive with over 20\% training samples. As seen, this strategy results in slightly better performance than setting a fixed value for all the primitives. However, in our current version, we directly set $K=5$ for all the primitives for simplicity. This experiment is added in Appendix Sec.D and Table 11. Thanks.
>
> | *K* range | Seen | Unseen | HM | AUC |
> |:-----:|:-----:|:-----:|:-----:|:-----:|
> |unique value 5 | $70.7$ | $76.0$ | $58.5$ | $46.6$ |
> |$[1,5]$ | $\mathbf{71.0}$ | $\mathbf{76.0}$ | $\mathbf{58.6}$ | $\mathbf{46.8}$ |
>
> Also, as you noticed, it is interesting to explore how to automatically determine $K$ for different primitives. We found that there are some clustering techniques (like [ref1]), which can automatically determine the number of cluster centers.  However, after running their code, we find that their algorithms are complex and time-consuming. We consider this as the basis of our future work.  The discussions are incorporated into the Appendix Sec.G.
>
> [ref1] Deepdpm: Deep clustering with an unknown number of clusters. CVPR 2022.
>
>
> **Q2: Resource limitations on large datasets.**
>
> **A2**:  Sorry for this confusion. Unlike offline clustering [ref2] requiring recomputing sub-primitive prototypes over the entire dataset after each batch, our algorithm adopts **online clustering approach with momentum updates**, where prototypes are dynamically updated using the embeddings within the current batch (stated in L267-271). **This design ensures our approach is scalable to large datasets, with minimal and consistent resource overhead regardless of dataset size.**
>
> Also, we would like to clarify that we had provided the resource usage analysis below about UT-Zappos in Table 10. As seen, ClusPro hardly incurs any additional GPU memory overhead or trainable parameters, with only a slight training delay due to additional clustering loops. Note that the memory and clustering are discarded after training, and do not introduce extra overheads in inference.
>
>
> |Method|Trainable params|GPU memory|Training time per epoch|Inference speed|AUC|
> |:-----:|:-----:|:-----:|:-----:|:-----:|:-----:|
> |Baseline|8.7M| 18.2G| 4.0min| 14.6ms |41.0|
> |ClusPro (ours) |  $\mathbf{8.7M}$|  $\mathbf{18.5G}$| $\mathbf{4.6min}$| $\mathbf{14.6ms}$ |$\mathbf{46.6}$|
>
>
> [ref2] Deep clustering for unsupervised learning of visual features. ECCV 2018.
>
>
>
> ---
>
> We sincerely appreciate your thorough review and invaluable feedback, which have significantly contributed to the improvement of our work. We hope we have addressed all of your concerns. Please let us know if you require any additional information.

---

> ### Author Response · Authors · 2024-11-25
> **Looking forward to the discussion**
>
> Dear Reviewer,
>
> We sincerely appreciate the time and effort you have dedicated to reviewing our submission. We have submitted our rebuttal and would like to follow up to inquire whether our responses have sufficiently addressed your concerns.
>
> Please let us know if you have any remaining questions or require additional clarification. We value your feedback and are eager to ensure our work meets the highest standards.
>
> Authors

---

> > ### Comment · Reviewer_1rZT · 2024-11-28
> >
> > Thanks for the authors' response. My concerns have been addressed.

---

> > > ### Author Response · Authors · 2024-11-29
> > > **Response to Reviewer 1rZT**
> > >
> > > Dear Reviewer 1rZT,
> > >
> > > We sincerely appreciate your time and effort in reviewing our submission and providing valuable comments. If you have any further questions or suggestions, please feel free to share them.
> > >
> > > Sincerely yours,
> > >
> > > Authors.

---

### Official Review · Reviewer_siAz · 2024-11-05

**Soundness:** 3
**Presentation:** 3
**Contribution:** 3
**Rating:** 6
**Confidence:** 4

**Summary:**

This paper presents a clustering-based prototype mining framework namely CLUSPRO, for Compositional Zero-Shot Learning.
Specifically, it considers within-primitive online clustering for automatically discovering and dynamically
updating prototypes. Besides, it models prototype-based primitive representation learning for promoting intra-primitive separation and inter-primitive decorrelation. Experimental results on three datasets demonstrate the superiority of the proposed method against existing methods.

**Strengths:**

Overall, the paper is well written and easy to follow.
The motivation behind addressing CZSL is clear and convincing.
The technical contributions are solid and novel.
The algorithm details are well provided and it should be possible to reproduce the results reported in the paper.

**Weaknesses:**

Figure 2 shows  the clustering-based prototype mining framework, while it is still unclear about the whole pipeline for CZSL. It is encouraged to introduce the complete network architecture in the main paper or supplementary material. It will be help to understand why the proposed method can achieve new state-of-the-art performance.

In the experiments, it is concerned about why all the compared methods including CLUSPRO employ the same ViT backbone,
to avoid the unfair comparisons between ViT-B and ViT-L.

It is interesting to know more details about the baseline model, including network structure and training loss.

**Questions:**

see the weaknesses above please.

---

> ### Author Response · Authors · 2024-11-22
> **Point-to-Point Response to Reviewer siAz**
>
> We thank reviewer siAz for the valuable time and constructive feedback. We provide point-to-point response below.
>
> ---
>
>
>
> **Q1: Clarification of the whole pipeline.**
>
>
>
>
>
> **A1**: Thanks for your suggestion. We have further provided the complete pipeline below of our algorithm in Appendix Sec.B to make readers fully understand our framework.  Thank you.
>
> >Fig. 2 presents the architecture of our ClusPro. It takes a batch of images and all the semantic labels (*i.e.*, attributes, objects, and compositions) as input. ClusPro first utilizes the visual encoder of CLIP [38] along with attribute and object adapters to obtain attribute features $F^a$, object features $F^o$, and composition features $F$ (Eq. 1). Besides, ClusPro constructs attribute, object, and composition prompt representations (Eq. 2) via a soft learnable prompt strategy [14] based on pre-given semantic labels. $\cdots\cdots$ Finally, we assemble the three-path classification loss $\mathcal{L}^{\text{BAS}}$ and our proposed prototype-based loss constraints (*i.e.*, $\mathcal{L}^{\text{PCL}}$ and $\mathcal{L}^{\text{PDL}}$)  as our final learning objective.
>
>
> ---
>
>
>
> **Q2: Comparison with the same backbone.**
>
>
>
> **A2**: Apologies for missing the backbone information; we have updated it in Table 1 and 2.
>
>
> |  Method |Backbone| MIT-States$_{AUC}$ | UT-zappos$_{AUC}$ | CGQA$_{AUC}$ |
> |:-----:|:-----:|:-----:|:-----:|:-----:|
> |CLIP  |ViT-L| 11.0|  5.0 | 1.4|
> |CoOp |ViT-L| 13.5|  18.8 | 4.4|
> |PCVL  |ViT-L |18.3  |32.2 |-|
> |CSP  |ViT-L |19.4 | 33.0|  6.2|
> |DFSP(i2t)| ViT-L  |20.7|  32.1 | 8.7|
> |DFSP(BiF) |ViT-L |  20.8 | 33.5  |9.9|
> |DFSP(t2i) |ViT-L|  20.6|  36.0 | 10.5|
> |GIPCOL |ViT-L|  19.9 | 36.2  |7.1|
> |CDS-CZSL |ViT-L | 22.4 | 39.5 | 11.1|
> |Troika  |ViT-L |22.1  |41.7  |12.4|
> |PLID |ViT-L  |22.1 | 38.7|  11.0|
> |CLUSPRO (Ours)|  ViT-L| $\mathbf{23.8}$  |$\mathbf{46.6}$  |$\mathbf{14.9}$|
>
> As seen, **all the compared methods including our approach employ the same ViT-L backbone**. Based on the above, we believe we render a fair comparison to existing work. Thanks.
>
>
>
> ---
>
>
>
> **Q3: More details about the baseline model.**
>
>
>
> **A3**: Following your suggestion, we provide more details below about the baseline model in Appendix Sec.C for clarity, as:
>
>  >**Visual Feature Extraction.** Following [14, 15, 23], we adopt the visual encoder ${\phi}^{\text{vis}}$ of CLIP [38] to splits the input image $X\in\mathbb{R}^{H \times W \times 3}$ into $N_{p}=HW/P$ patches, where $P$ is the resolution of each patch. $\cdots\cdots$  We adopt attribute adapter ${h}^{a}$ and object adapter ${h}^{o}$, each implemented as a separate MLP, to project ${f}^{c}$ into the discriminative attribute feature ${f}^{a}$ and object feature ${f}^{o}$, respectively.
> **Prompt Feature Extraction.**  We follow existing CZSL [18, 14] to employ an independent prompt prefix for each branch.  $\cdots\cdots$ Then these prompts are then fed into the frozen text encoder of CLIP [38] to obtain prompt features.
> **Training Loss $\mathcal{L}^{\text{BAS}}$.** Following previous CZSL approaches [14, 15, 23], the parameters $\theta$ of the baseline model are learned by minimizing the three-path classification loss (Eq.14)  on the training dataset. Note that we have omitted the weight decay in Eq.14 for simplicity. We just follow [14] set the weight decay as 5e−5 for all our experiments.
> **Feasibility Calibration for Open-World Setting**. Following [17, 14], we adopt post-training feasibility calibration to filter out infeasible compositions that might be present in the open-world setting. The calibration relies on the assumption that similar objects tend to share similar attributes, while dissimilar objects are unlikely to exhibit shared attributes.$\cdots\cdots$
>
>
> ---
>
>
>
> We appreciate again your thoughtful review and we hope we addressed your concerns. Please let us know if you'd like any further information.

---

> > ### Comment · Reviewer_siAz · 2024-11-26
> >
> > Thanks for the rebuttal which has addressed my concerns clearly.

---

> > > ### Author Response · Authors · 2024-11-27
> > > **Response to Reviewer siAz**
> > >
> > > Dear Reviewer siAz,
> > >
> > > We greatly appreciate your valuable time and constructive feedback. If you have any further questions or suggestions, please feel free to share them.
> > >
> > > Sincerely yours,
> > >
> > > Authors

---

> ### Author Response · Authors · 2024-11-25
> **Looking forward to the discussion**
>
> Dear Reviewer,
>
> We sincerely appreciate the time and effort you have dedicated to reviewing our submission. We have submitted our rebuttal and would like to follow up to inquire whether our responses have sufficiently addressed your concerns.
>
> Please let us know if you have any remaining questions or require additional clarification. We value your feedback and are eager to ensure our work meets the highest standards.
>
> Authors

---

### Author Response · Authors · 2024-11-22
**Summary of Revisions**

We express our sincere gratitude to all reviewers for their valuable time and thorough assessment of our manuscript. We have revised our paper according to your comments (where the revised parts are highlighted in blue color). The major changes are as follows:

1. We clarify the complete pipeline of our framework, according to Reviewer siAz's suggestion in Appendix Sec.B.
2. We add more details about the baseline model, according to Reviewer siAz's suggestion in Appendix Sec.C.
3. We add the backbone information, according to Reviewer siAz's suggestion in Table 1 and 2.
4. We study the performance when varying the number of prototypes with the number of training samples for each primitive, according to Reviewer 1rZT's comments in Appendix Sec.D, Table 11.
5. We offer more detailed discussions regarding automatically determining K for different primitives, according to Reviewer 1rZT's comments in Appendix Sec.G.
6. We add experiments to evaluate the performance of our model when not utilizing pre-trained CLIP model, according to Reviewer Aocv's comments in Appendix Sec.D, Table 9.
7. We add detailed data overlap discussion about CLIP, according to Reviewer Aocv's comments in Appendix Sec.G.
8. We discuss the related work on prototype learning in ZSL/CZSL and the difference between ours and previous works, according to Reviewer Q4uv's comments.
9. We improve the presentation of Sec. 3.3, according to Reviewer Q4uv's comments.
10. We discuss the novelty of our negative sample selection in prototype-based contrastive learning, according to Reviewer Q4uv's comments in Sec. 3.4.


Please refer to our response for more details. We have strived to address each of your concerns and welcome further discussions and insights.

Sincerely yours,

Authors.

---

### Meta-Review · Area_Chair_FPrq · 2024-12-22

**Metareview:**

In this paper, the CLUSPRO framework is introduced for Compositional Zero-Shot Learning, utilizing clustering-based prototype mining. It incorporates within-primitive online clustering to automatically find and update prototypes dynamically. Additionally, it employs prototype-based primitive representation learning to enhance both intra-primitive separation and inter-primitive decorrelation. Results from experiments on three datasets showcase the effectiveness of this approach compared to existing methods. The AC considers this approach as interesting and effective. The results also support the claims of the paper.

**Additional Comments On Reviewer Discussion:**

After the rebuttal, all reviewers lean towards the positive side. The AC did not spot any obvious reasons to turn down the reviewers' recommendation. Therefore, the AC would like to recommend for acceptance. Authors please revise the final paper incorporating the required revision raised by the reviewers. Meanwhile, a careful discussion on the potential bias caused by the use of the pretrained large VL model like CLIP should be included, in response to the potential ethical issues.

---

### Decision · Program_Chairs · 2025-01-22

Accept (Poster)